# Mitigating the non-specific uptake of immunomagnetic microparticles enables the extraction of endothelium from human fat

Jeremy A. Antonyshyn [1,2], Vienna Mazzoli [1,2], Meghan J. McFadden[1,2], Anthony O. Gramolini [2,3], Stefan O. P. Hofer[4,5], Craig A. Simmons[1,2,6] & J. Paul Santerre [1,2,7 ✉]

Endothelial cells are among the fundamental building blocks for vascular tissue engineering. However, a clinically viable source of endothelium has continued to elude the field. Here, we demonstrate the feasibility of sourcing autologous endothelium from human fat – an abundant and uniquely dispensable tissue that can be readily harvested with minimally invasive procedures. We investigate the challenges underlying the overgrowth of human adipose tissue-derived microvascular endothelial cells by stromal cells to facilitate the development of a reliable method for their acquisition. Magnet-assisted cell sorting strategies are established to mitigate the non-specific uptake of immunomagnetic microparticles, enabling the enrichment of endothelial cells to purities that prevent their overgrowth by stromal cells. This work delineates a reliable method for acquiring human adipose tissue-derived microvascular endothelial cells in large quantities with high purities that can be readily applied in future vascular tissue engineering applications.

[1] Institute of Biomedical Engineering, University of Toronto, Toronto, ON, Canada. [2] Translational Biology and Engineering Program, Ted Rogers Centre for Heart Research, Toronto, ON, Canada. [3] Department of Physiology, University of Toronto, Toronto, ON, Canada. [4] Division of Plastic, Reconstructive, and Aesthetic Surgery, University of Toronto, Toronto, ON, Canada. [5] Departments of Surgery and Surgical Oncology, University Health Network, Toronto, ON, Canada. [6] Department of Mechanical and Industrial Engineering, University of Toronto, Toronto, ON, Canada. [7] Faculty of Dentistry, University of Toronto, Toronto, ON, Canada. ✉email: paul.santerre@utoronto.ca

Comprising a single layer of cells lining the luminal surface of the vasculature, the endothelium is the interface between blood and the different tissues of the body[1]. Although the endothelium exhibits marked phenotypic heterogeneity between vascular beds, its fundamental functions are to maintain blood in a fluid state and to confine its flow to the boundaries of the vasculature[1]. Endothelial cells (ECs) are consequently indispensable for vascular tissue engineering, enabling the vascularization and sustained perfusion of engineered tissues[2], as well as the endothelialization and prolonged patency of small-diameter vascular prostheses[3]. Despite being the fundamental building block for vascular tissue engineering, a clinically viable source of endothelium has continued to elude the field[4].

The ideal source of ECs for vascular tissue engineering will be autologous to preclude immunogenic concerns and readily accessible to minimize the time needed for their culture-mediated expansion[3]. The scarcity of expendable blood vessels and the low prevalence of microvascular ECs in tissues has prompted many to turn to alternative sources of endothelium, namely stem and progenitor cells[5–7]. However, their inherent proliferative and differentiative potential, coupled with the spectrum of manipulations used to instill in them an endothelial phenotype, introduces distinct regulatory concerns that may impede their clinical translation when compared with their natively differentiated and minimally manipulated counterparts[8,9]. Remarkably, an abundant and uniquely dispensable source of autologous endothelium remains largely untapped: adipose tissue.

Adipose tissue is an attractive source of ECs for vascular tissue engineering because it can be harvested autologously in large quantities with minimally invasive procedures using a cannula coupled to a liposuction pump or even a syringe[10]. While the need for the culture-mediated expansion of human adipose tissue-derived microvascular ECs (HAMVECs) is mitigated by the abundant and uniquely dispensable nature of the tissue, their low prevalence has continued to complicate their acquisition[3,11–15]. Specifically, their primary cultures are often overgrown by residual stromal cells from the cell sorting procedure[3,11–15]. This challenge in acquiring HAMVECs may account for their strikingly limited adoption for vascular tissue engineering[16–18]. Moreover, the absence of an accessible alternative likely contributes to the sustained popularity of human umbilical vein ECs (HUVECs), the use of which continues to be reported in ~ 60% of publications amongst the biomaterials and tissue engineering communities[4,19] despite being an allogeneic and, thereby, clinically impractical source of endothelium[20]. Accordingly, there is a clear and unmet need for a readily accessible and non-immunogenic source of endothelium if tissue-engineered constructs are to leave the bench for the bedside.

The overgrowth of microvascular ECs by residual stromal cells from the cell sorting procedure is a common complication. This phenomenon is not specific to HAMVECs[3,11–15], but challenges the isolation of microvascular ECs from nearly all tissues, including the brain[21], kidney[22], and lung[23]. While early attempts at their isolation relied on sieves and differential centrifugation to enrich the endothelium from enzymatically digested tissues[24], these approaches have been largely replaced or augmented with immunoselection, which allows for the enrichment of the endothelium based on its unique cell-surface protein signature[25]. Nevertheless, stromal cell contamination has continued to complicate their primary cultures, driving many to resort to differential adhesion[11,15,21], clonal selection[23], and manual weeding[12,22] in order to prevent their impendent overgrowth. These procedures are labour-intensive, time-consuming, and of uncertain reproducibility, comprising a formidable obstacle to their widespread adoption for vascular tissue engineering.

The objective of this study was to develop an accessible and reliable method of acquiring endothelium from human fat for vascular tissue engineering. Magnet-assisted cell sorting (MACS) was explored for the immunoselection of HAMVECs due to its low cost, small physical footprint, and ease of use. Their overgrowth by stromal cells was first investigated to gain a clear understanding of the specific challenges underlying this phenomenon, which was then used to inform the development of a reproducible method of acquiring HAMVECs. Strategies were established to mitigate the non-specific uptake of immunomagnetic microparticles (IMPs), enabling the enrichment of HAMVECs to purities that prevent their overgrowth by stromal cells. This study demonstrates the feasibility of sourcing autologous endothelium from human fat for vascular tissue engineering. It delineates a reliable and facile method for its acquisition that can be readily implemented by the field.

## Results

**HAMVECs were often overgrown by residual ASCs from the cell sorting procedure**. HAMVECs were isolated from the stromal vascular fraction of enzymatically digested human subcutaneous abdominal white adipose tissue using MACS (Fig. 1). The yield of stromal vascular cells was $6.6 \pm 4.7 \times 10^5$ cells per gram of tissue. HAMVECs were extracted from the stromal vascular fraction on the basis of a CD45$^-$CD31$^+$ immunophenotype—i.e. a cell-surface protein signature characteristic of differentiated endothelium[26–28]. The stromal vascular fraction was first depleted of CD45$^+$ leucocytes prior to positively selecting for CD31$^+$ HAMVECs due to the high prevalence of leucocytes ($41.8 \pm 4.7\%$ CD45$^+$; Fig. 1a; Supplementary Fig. 1) and their capacity to co-express this characteristic endothelial cell-surface marker ($43.0 \pm 3.2\%$ CD31$^+$; Supplementary Fig. 2).

The putative CD45$^-$CD31$^+$ HAMVECs comprised $0.9 \pm 0.6\%$ of the stromal vascular fraction (Fig. 1a), which is comparable to the prevalence of microvascular ECs in other tissues[25]. Their yield was $4.6 \pm 3.0 \times 10^3$ cells per gram of fat. Cultures of HAMVECs were significantly enriched for the CD45$^-$CD31$^+$ immunophenotype when compared with the stromal vascular fraction ($98.6 \pm 0.9\%$ vs. $0.9 \pm 0.6\%$, respectively; $p < 0.0001$; Fig. 1b), and they exhibited a characteristic endothelial cobblestone-like morphology (Fig. 1c). Cultures of these purities ($98.6 \pm 0.9\%$ CD45$^-$CD31$^+$; range: 98.0–99.7% CD45$^-$CD31$^+$) were successfully established from three patients to enable the ensuing investigations of the challenges underlying their acquisition.

The isolation of HAMVECs had to be attempted from 20 patients in order to obtain these three cultures of a characteristic endothelial cobblestone-like morphology. In the other 17 patients, HAMVECs were visibly overgrown within 2 weeks by spindle-shaped, fibroblast-like cells (Fig. 1g). These contaminating cells comprised residual CD45$^-$CD31$^-$ stromal vascular cells from the MACS procedure (Fig. 1f). Importantly, sequential enrichments for the CD31$^+$ immunophenotype failed to eliminate these contaminating cells and prevent their overgrowth of HAMVECs (Fig. 1g).

CD45$^-$CD31$^-$ stromal vascular cells were retained from subsequent isolations to facilitate investigations of their overgrowth of HAMVECs. They comprised $57.2 \pm 5.0\%$ of the stromal vascular fraction (Fig. 1a), and their yield was $3.8 \pm 2.7 \times 10^5$ cells per gram of tissue. Their cultures were significantly enriched for the CD45$^-$CD31$^-$ immunophenotype when compared with the stromal vascular fraction ($99.8 \pm 0.2\%$ vs. $57.2 \pm 5.0\%$, respectively; $p = 0.0019$; Fig. 1d), and they exhibited a comparable morphology to the contaminating cells (Fig. 1e). Notably, these CD45$^-$CD31$^-$ stromal vascular cells comprised adipose tissue-derived stromal/stem cells (ASCs)[29], having been previously

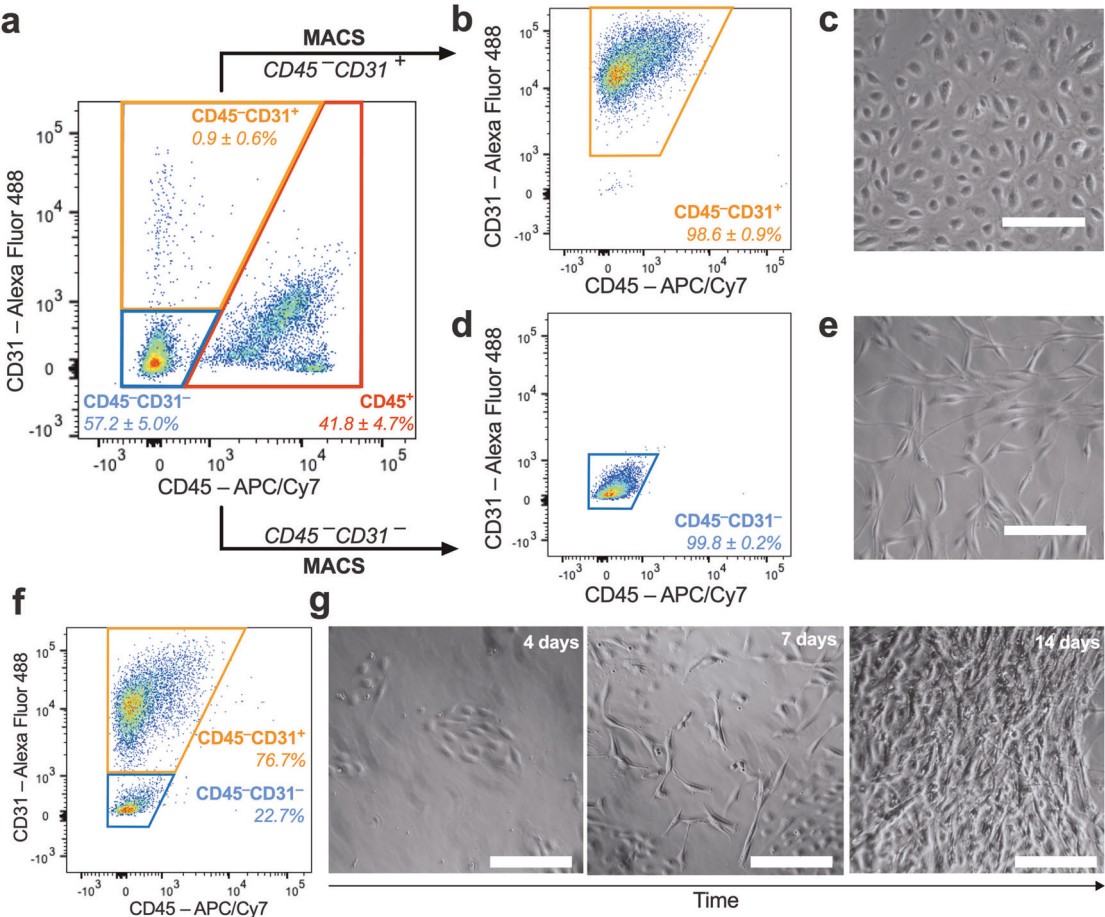

**Fig. 1 Primary cultures of human adipose tissue-derived microvascular endothelial cells (HAMVECs) are often overgrown by residual adipose tissue-derived stromal/stem cells (ASCs) from the magnet-assisted cell sorting (MACS) procedure.** The stromal vascular fraction of enzymatically digested human subcutaneous abdominal white adipose tissue (**a**) was depleted of CD45$^+$ leucocytes prior to positively selecting for CD31 expression to establish primary cultures of CD45$^-$CD31$^+$ HAMVECs (**b**, **c**). Their primary cultures were often overgrown by residual CD45$^-$CD31$^-$ ASCs from the MACS procedure despite sequential enrichments for CD31 expression (**f**, **g**), prompting the retention of ASCs for downstream studies (**d**, **e**). Shown are representative pseudocolour plots depicting the composition of the different populations of cells (**a**, **b**, **d**, **f**), as well as representative photomicrographs depicting their corresponding morphologies (**c**, **e**, **g**). Scale bars represent 200 μm; values mean ± standard deviation. While the isolation of HAMVECs was initially attempted from twenty patients ($n = 20$ biologically independent samples), only three cultures were visibly free of stromal cell overgrowth ($n = 3$ biologically independent samples); in the other 17 patients, cultures were visibly overgrown by stromal cells within 2 weeks ($n = 17$ biologically independent samples). The composition of the different populations of cells was assessed in three patients in all but visibly contaminated primary cultures of HAMVECs (**a**, **b**, **d**; $n = 3$ biologically independent samples), in which flow cytometry was used to elucidate the identity of the contaminating stromal cells rather than to assess their purity (**f**; $n = 1$ biologically independent sample).

validated to meet the phenotypic criteria delineated by the *International Federation for Adipose Therapeutics and Science* and the *International Society for Cellular Therapy*[30].

**HAMVECs were committed to the endothelial lineage**. The phenotypic heterogeneity of the endothelium and the paucity of markers with specificity and sensitivity for the endothelial lineage presents a challenge to the assessment of an endothelial phenotype[1]. It has previously misled many to mistake monocytes for endothelial progenitor cells[31], platelets for circulating ECs[32], omental mesothelial cells for HAMVECs[33,34], and even ASCs for EC substitutes[29]. Accordingly, comprehensive phenotypic comparisons to representative EC controls were performed to confirm the endothelial phenotype of the putative HAMVECs. Specifically, HAMVECs were compared to controls representative of the predominant endothelial specializations[35], namely arterial (control: human coronary artery ECs, HCAECs) vs. venous (control: HUVECs) and macrovascular (controls: HCAECs and HUVECs)

vs. microvascular (control: human dermal microvascular ECs, HDMVECs).

The endothelial phenotype of the putative HAMVECs was first validated using a targeted assessment of characteristic endothelial traits (Fig. 2). HAMVECs shared a similar cobblestone-like morphology with the HUVECs, HCAECs, and HDMVECs (Fig. 2a). The abundance of transcripts encoding CD31 (gene: *PECAM1*), vascular endothelial (VE)-cadherin (gene: *CDH5*), and von Willebrand Factor (vWF; gene: *VWF*) in HAMVECs was statistically equivalent to that observed in the EC controls (Fig. 2b; Supplementary Table 1; and Supplementary Fig. 3), and they exhibited comparable expression of the corresponding proteins (Fig. 2c). Furthermore, their uptake of acetylated low-density lipoprotein (AcLDL) was similar (Fig. 2d). Importantly, these morphological, molecular, and functional endothelial hallmarks exhibited by HAMVECs were previously shown to be negligible in ASCs when cultured under identical conditions and compared with the same EC controls[29]. Lastly, HAMVECs exhibited an angiogenic capacity comparable to that of the EC

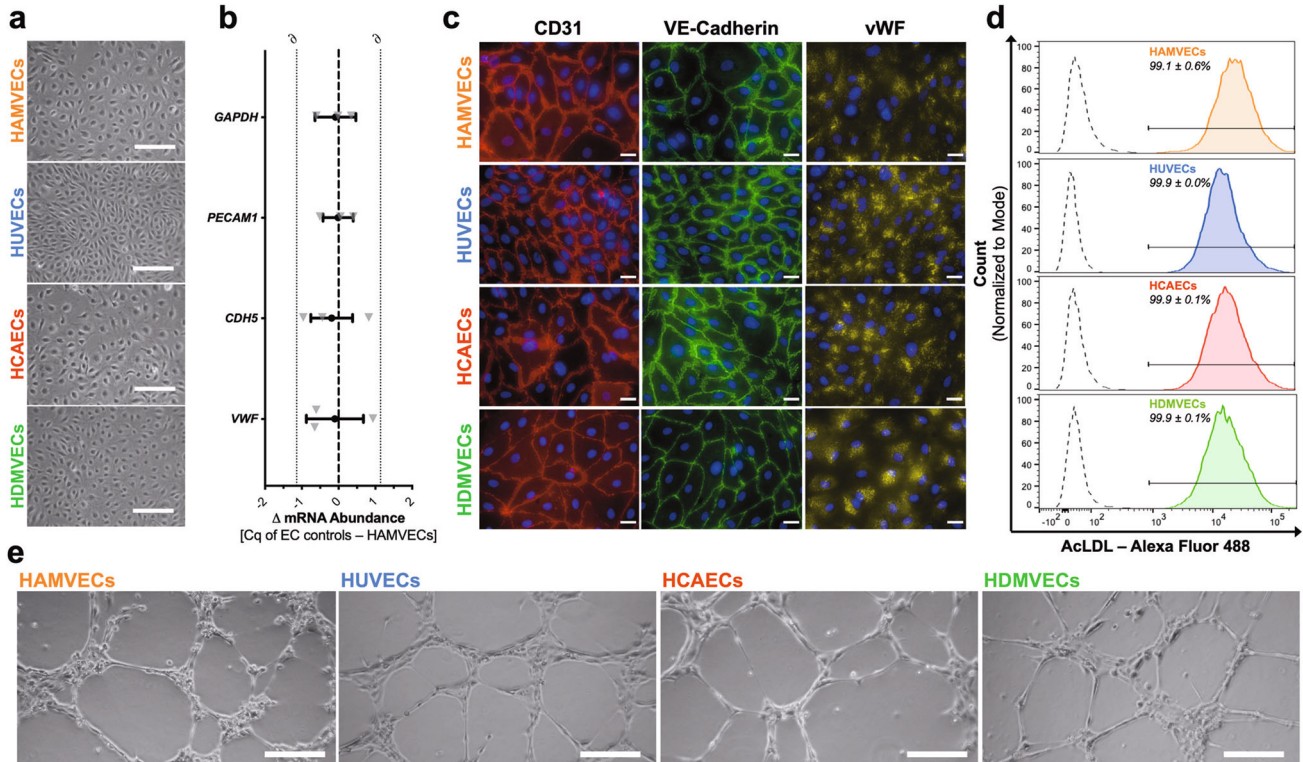

**Fig. 2 Human adipose tissue-derived microvascular endothelial cells (HAMVECs) exhibit morphological, molecular, and functional hallmarks of endothelium.** HAMVECs were compared with endothelial cell (EC) controls representative of the predominant endothelial specializations, namely human umbilical vein ECs (HUVECs; macrovascular, venous endothelium), human coronary artery ECs (HCAECs; macrovascular, arterial endothelium), and human dermal microvascular ECs (HDMVECs; microvascular endothelium). **a** Cobblestone-like morphology of endothelium. Scale bars represent 200 μm. **b** Abundance of transcripts encoding CD31 (gene: *PECAM1*), vascular endothelial (VE)-cadherin (gene: *CDH5*), and von Willebrand Factor (vWF; gene: *VWF*). Glyceraldehyde-3-phosphate dehydrogenase (gene: *GAPDH*) was used as a loading control. Dashed line depicts a mean difference of zero; and dotted lines is the equivalence margin (∂) used for the two one-sided test for equivalence. Values represent the mean ± 90% confidence interval; mRNA, messenger ribonucleic acid; and $C_q$, the quantification cycle. **c** Expression and localization of the corresponding endothelial proteins. Scale bars represent 25 μm. **d** Internalization of acetylated low-density lipoprotein (AcLDL). Solid and dashed lines represent ECs cultured in the presence and absence of Alexa Fluor 488-conjugated AcLDL, respectively. Values represent mean ± standard deviation. **e** Capillary-like tubulogenesis by ECs. Scale bars represent 200 μm. All experiments were performed in biological triplicate, using cells derived from three different donors ($n = 3$ biologically independent samples).

controls (Fig. 2e), further supporting their functional endothelial phenotype.

The commitment of HAMVECs to the endothelial lineage was then assessed. Specifically, their adipogenic plasticity was compared to that of ASCs (Fig. 3). While the culture of ASCs in adipogenic media increased the size of their lipid droplets (Fig. 3a), it did not significantly increase their total abundance of lipids (Fig. 3b). This discrepancy may be attributed to differences in their cellular densities, as the growth of ASCs was halted at the confluence in adipogenic media but not in endothelial media (Fig. 3a). Nevertheless, ASCs exhibited a significantly greater accumulation of lipids than HAMVECs regardless of the media in which they were cultured (Fig. 3b). Furthermore, the culture of HAMVECs in adipogenic media induced their death rather than adipogenesis (Fig. 3a), supporting that HAMVECs are fully differentiated and committed to the endothelial lineage.

**Proteomic assessment of HAMVECs implicated heterotypic cell–cell interactions in the modulation of their overgrowth by ASCs.** The phenotype of HAMVECs was further compared to that of the representative EC controls using liquid chromatography–tandem mass spectrometry (LC–MS/MS; Fig. 4). While the endothelial proteome was 87% conserved (Fig. 4c), unsupervised hierarchical clustering of their global proteomes

effectively grouped ≥ 2 biological replicates from each of the four different vascular beds (Fig. 4b), supporting that the phenotypic heterogeneity of endothelium persists despite its culture under identical conditions[35]. HAMVECs exhibited the most distinctive proteome (Fig. 4b, d), with 284 proteins differentiating them from all other endothelial specializations. Importantly, gene ontological analyses did not identify any biological pathways related to angiogenesis, haemostasis, nor permeability, that were enriched amongst these differentially expressed proteins, adding further support to the endothelial phenotype of the CD45⁻CD31⁺ stromal vascular cells. Rather, pathways implicated in proliferation were found to be statistically overrepresented (Fig. 4e), suggesting that the proliferative capacity of HAMVECs was their most differentiating feature.

HAMVECs exhibited a significantly longer population doubling time than HUVECs, HCAECs, and HDMVECs (Fig. 4f). Interestingly, their slower proliferation was associated with the slight but significantly lower purity of their cultures (Fig. 4g), suggesting that residual ASCs from the cell sorting procedure may suppress the proliferation of HAMVECs to facilitate their overgrowth of primary cultures. However, HAMVECs could be repeatedly sub-cultured and maintained at confluence for >3 weeks without any signs of stromal cell overgrowth despite their trace impurities. In fact, zones of inhibition appeared to

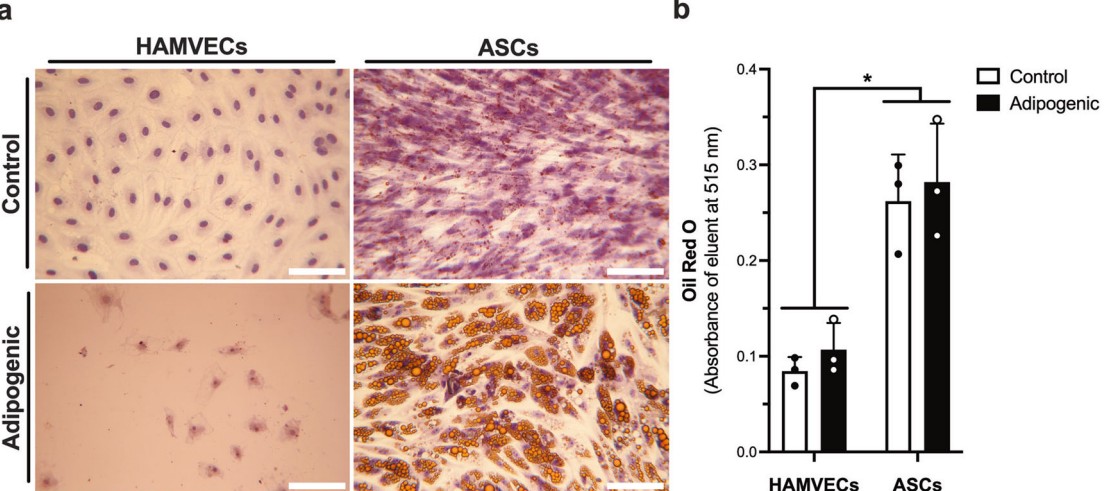

**Fig. 3 Adipogenic plasticity is evident in adipose tissue-derived stromal/stem cells (ASCs), not human adipose tissue-derived microvascular endothelial cells (HAMVECs).** HAMVECs and ASCs were cultured in adipogenic medium for 10 days before their accumulation of lipids was assessed by Oil Red O with hematoxylin counterstaining. Endothelial medium is used as control. Shown are representative photomicrographs depicting the accumulation of lipids by HAMVECs and ASCs (**a**), as well as a bar graph delineating its quantification (**b**). Hematoxylin stains nuclei blue/purple, and Oil Red O stains lipids red/orange. Scale bars represent 100 μm. Values represent mean ± standard deviation; and *$p < 0.05$. Experiments were performed in biological triplicate, using cells derived from three different donors ($n = 3$ biologically independent samples).

surround the residual stromal cells in these cultures (Fig. 4h), suggesting that both HAMVECs and ASCs are capable of suppressing each other's proliferation.

HAMVECs and ASCs were seeded in pre-defined proportions to recapitulate different efficacies of enrichment for the $CD45^-CD31^+$ immunophenotype, and the effect of seeding purity on their proliferation and the consequent temporal composition of their cultures was assessed (Fig. 5). HAMVECs exhibited a significantly longer population doubling time than ASCs (Fig. 5a), and their population growth rate was further suppressed with lower seeding purities (Fig. 5b; Supplementary Fig. 4). HAMVECs were also found to inhibit the growth of ASCs, albeit to a significantly lower extent (Fig. 5b). Although similar magnitudes of population growth inhibition in both HAMVECs and ASCs were observed with a 90% seeding purity (Fig. 5b), the longer population doubling time of HAMVECs (Fig. 5a) suggests that an enrichment efficacy >90% is required in order for the absolute population growth rate of HAMVECs to exceed that of ASCs—i.e. the threshold of purity beyond which stromal cell overgrowth is precluded. In fact, cultures of HAMVECs established with 90% purity consistently exhibited significant overgrowth by ASCs within 7 days, and HAMVECs were virtually undetectable in these cultures after 28 days (Fig. 5c). In contrast, our isolates ($98.6 \pm 0.9\%$ $CD45^-CD31^+$; range: 98.0–99.7% $CD45^-CD31^+$) could be maintained in culture for 28 days without a significant decline in purity (Fig. 5c). These findings underscore the importance of seeding purity to the protracted stability of HAMVEC cultures, and suggest that no further enrichment is needed in cultures established with purities ≥98%.

**ASCs exhibited a capacity to bind and internalize anti-CD31 IMPs.** The failure of sequential rounds of MACS to eliminate contaminating ASCs from primary cultures suggests that they are capable of binding the anti-CD31 IMPs. Interestingly, ASCs exhibited both a membrane-bound and intracellular localization of the anti-CD31 IMPs (Fig. 6a), indicating that they can further internalise them. The prevalence of ASCs exhibiting a membrane-bound or intracellular localization of anti-CD31 IMPs after 20 min in suspension (i.e. labelling conditions for MACS) was $17.1 \pm 3.3\%$ (Fig. 6b; Supplementary Fig. 5), supporting a role for

the IMPs in mediating the contamination of primary cultures. While the prevalence of membrane-bound anti-CD31 IMPs decreased over time in culture, the prevalence of their internalization significantly increased (Fig. 6b; $p < 0.0001$). In fact, the proportion of ASCs exhibiting bound and internalized anti-CD31 IMPs was significantly greater after 48 h in culture than after 20 mins in suspension (Fig. 6c), accounting for the failure of sequential rounds of MACS to eliminate the ASCs. Moreover, IMP-laden ASCs exhibited comparable levels of deoxyribonucleic acid (DNA) synthesis as IMP-free ASCs (Fig. 6d; Supplementary Fig. 6), suggesting that their binding and internalization of the anti-CD31 IMPs does not significantly affect their capacity to proliferate and overtake HAMVECs following their magnetic separation and sub-culture.

**Uptake of anti-CD31 IMPs by ASCs was non-specific, but size and exposure -dependent.** The capacity of ASCs to express low levels of characteristic endothelial markers is well-documented[29], suggesting that their binding and internalization of the anti-CD31 IMPs may be a manifestation of their limited endothelial plasticity. LC–MS/MS was used to identify an alternative immunophenotypic marker with greater specificity for HAMVECs in cultures contaminated with ASCs (Fig. 7). While unsupervised hierarchical clustering of their global proteomes supported the distinct phenotypes of HAMVECs and ASCs (Fig. 7b), 88% of their proteomes were conserved (Fig. 7c). HAMVECs were found to express all of the characteristic immunophenotypic markers traditionally used to define ASCs[30], including CD90, CD44, CD13, CD73, CD29, and CD105 (Fig. 7d). Furthermore, the endothelial plasticity of the ASCs was evident, with LC–MS/MS detecting their expression of CD31, VE-cadherin, and vWF, albeit in significantly lower abundances when compared with the HAMVECs (Fig. 7d). These findings suggest that HAMVECs cannot be enriched, nor can ASCs be depleted, from primary cultures on the basis of their conventional immunophenotypes— including CD31. Gene ontological mapping of the 457 proteins enriched in HAMVECs to their cellular components identified 188 that were localized to the plasma membrane (Supplementary Fig. 7), of which 37 were detected with specificity and sensitivity (i.e. $n = 3/3$ HAMVECs and $n = 0/3$ ASCs). CD93 was selected as

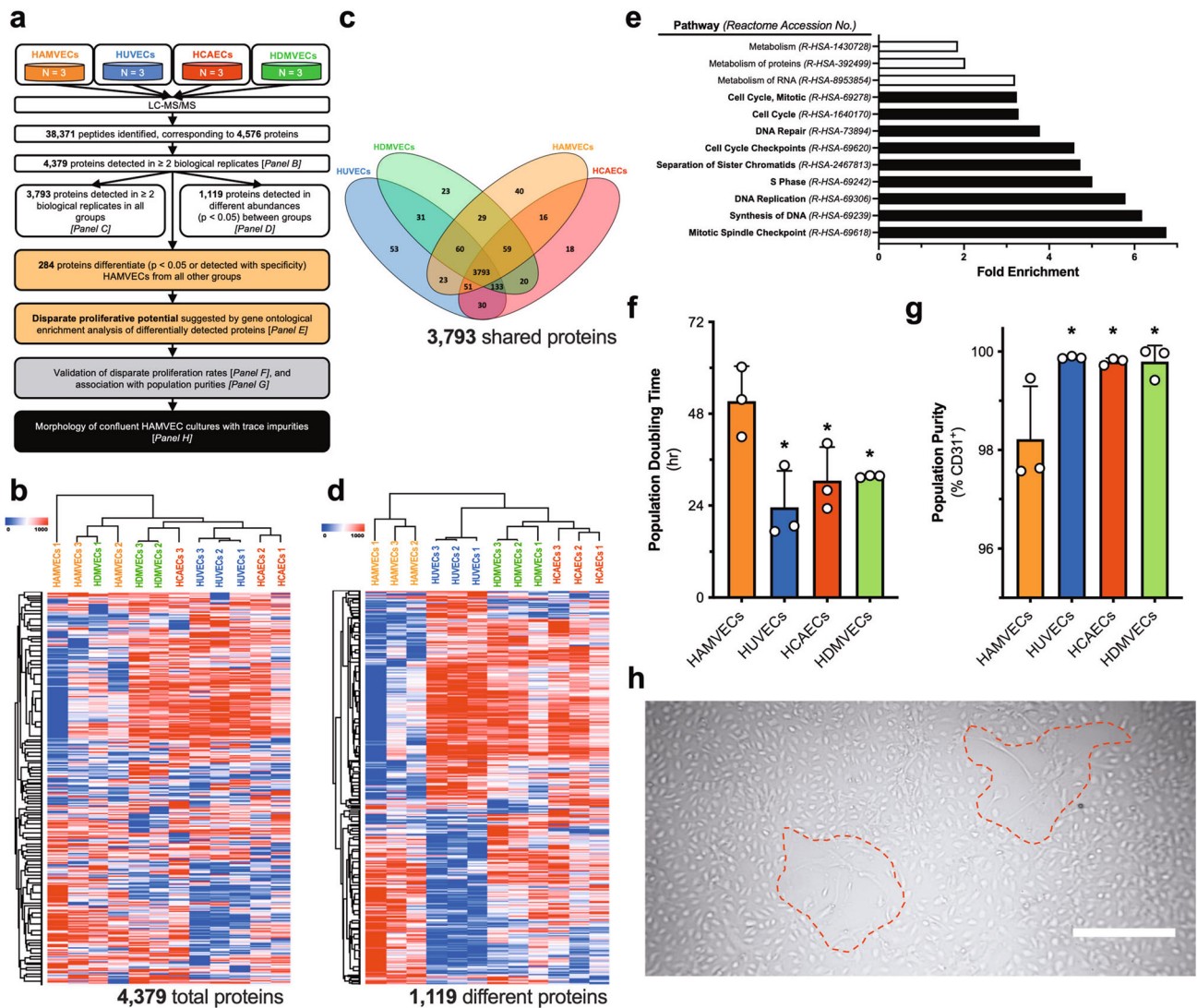

**Fig. 4 Proteomic assessment of human adipose tissue-derived microvascular endothelial cells (HAMVECs) suggests that their proliferation is suppressed by adipose tissue-derived stromal/stem cells (ASCs). a** Workflow depicting the proteomic comparison of HAMVECs with endothelial cell (EC) controls representative of the predominant endothelial specializations, namely human umbilical vein ECs (HUVECs; macrovascular, venous endothelium), human coronary artery ECs (HCAECs; macrovascular, arterial endothelium), and human dermal microvascular ECs (HDMVECs; microvascular endothelium). **b** Unsupervised hierarchical clustering of the proteomes of the ECs derived from four different vascular beds. **c** Distribution of detected proteins amongst the different ECs. **d** Hierarchical clustering of the proteins detected in different abundances ($p < 0.05$) between EC types. **e** Biological pathways differentiating HAMVECs from all other types of ECs. Black bars highlight those related to proliferation; and white bars metabolism. **f** Population doubling times of the different ECs, determined from their exponential growth phase observed over 7 days of culture. Values represent mean ± standard deviation; and *$p < 0.05$ compared with HAMVECs. **g** Purities of the different EC cultures. Values represent mean ± standard deviation; and, *$p < 0.05$ compared with HAMVECs. **h** Representative zones of inhibition surrounding residual ASCs in primary cultures of HAMVECs that were maintained at confluence for over 3 weeks. Scale bar represents 500 μm. All experiments were performed in biological triplicate, using cells derived from three different donors ($n = 3$ biologically independent samples).

a candidate immunophenotypic marker for HAMVECs due to its cell-surface localization and prior characterization[36].

The specificity and sensitivity of CD93 for HAMVECs was validated both in culture as well as in the freshly isolated stromal vascular fraction. While the sensitivity of CD93 for HAMVECs in vitro was slightly poorer than that of CD31 (HAMVECs: $99.6 \pm 0.4\%$ CD31$^+$ vs. $95.2 \pm 4.3\%$ CD93$^+$; $p = 0.2244$), its specificity was greater (ASCs: $8.4 \pm 5.8\%$ CD31$^+$ vs. $0.1 \pm 0.0\%$ CD93$^+$; $p = 0.1280$; Fig. 7e). Although these differences were not statistically significant and the detection of CD31 in ASCs required its secondary antibody-mediated signal amplification (Supplementary Fig. 8), these slight disparities in sensitivity and specificity may have meaningful implications on their capacities

to enrich for HAMVECs given the potent growth inhibition imposed by even a small number of residual ASCs. Interestingly, the specificity and sensitivity exhibited by CD93 for HAMVECs in vitro were not shared by HAMVECs in the stromal vascular fraction, with low levels of CD93 expression detected amongst CD45$^+$ leucocytes, CD45$^-$CD31$^-$ ASCs, and CD45$^-$CD31$^+$ HAMVECs (Fig. 7f). These findings suggest that although CD93 cannot be used to isolate HAMVECs from the stromal vascular fraction of enzymatically digested adipose tissue, it may be superior to CD31 for the sequential enrichment of their contaminated cultures.

The binding and internalization of anti-CD93 IMPs by ASCs was compared to that of the anti-CD31 IMPs. Anti-CD93

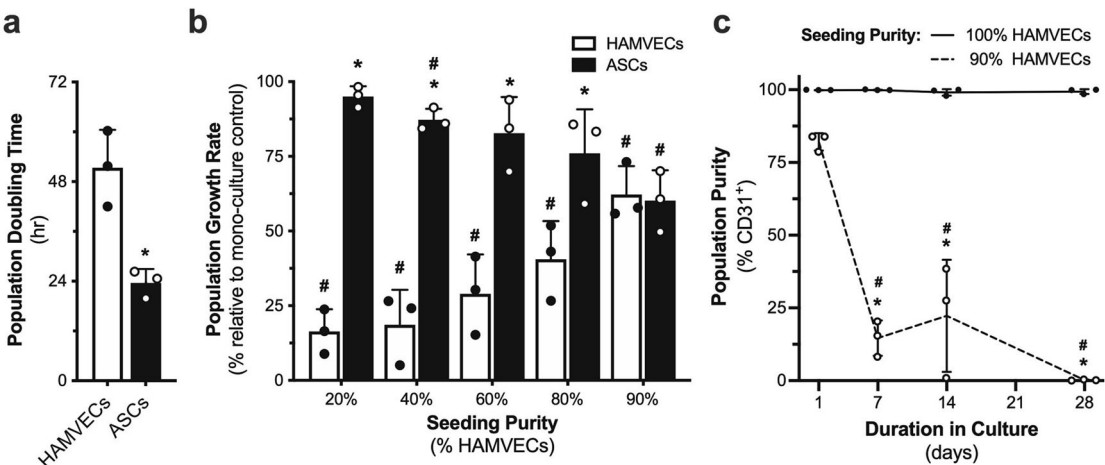

**Fig. 5 Heterotypic cell–cell interactions modulate the overgrowth of human adipose tissue-derived microvascular endothelial cells (HAMVECs) by adipose tissue-derived stromal/stem cells (ASCs). a** Population doubling times of HAMVECs and ASCs, determined from their exponential growth phase observed over 7 days of culture. *$p < 0.05$ compared with HAMVECs. **b** Effect of seeding purity on the population growth rates of HAMVECs and ASCs, observed over 4 days of culture. *$p < 0.05$ compared with HAMVECs; #$p < 0.05$ compared with 100%. **c** Effect of seeding purity on the composition of cultures maintained at confluence for up to 28 days. *$p < 0.05$ compared with 100% seeding purity at respective time-point; #$p < 0.05$ compared with purity at day 1. Values represent mean ± standard deviation. All experiments were performed in biological triplicate, using cells derived from three different donors ($n = 3$ biologically independent samples).

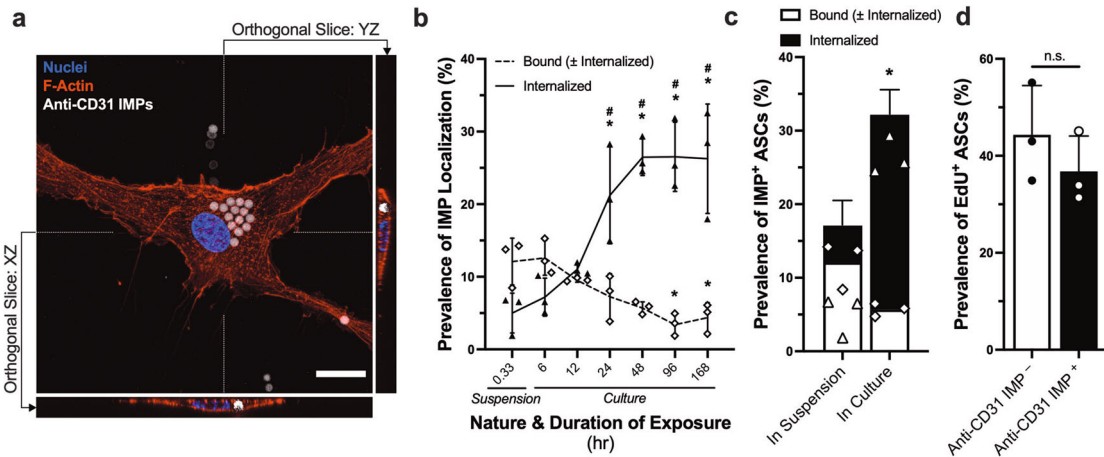

**Fig. 6 Adipose tissue-derived stromal/stem cells (ASCs) bind and internalize anti-CD31 immunomagnetic microparticles (IMPs). a** Intracellular localization of anti-CD31 IMPs in ASCs after 24 hr of culture. Scale bar represents 25 µm. **b** Prevalence of IMP localization in ASCs after 20 min in suspension (i.e. labelling conditions for magnet-assisted cell sorting procedure) and different durations in culture. *$p < 0.05$ compared with respective localization in suspension; #$p < 0.05$ compared with membrane-bound localization. **c** Total prevalence of ASC interactions with anti-CD31 IMPs after 20 min in suspension vs. 48 hr in culture. Shown is a stacked bar graph depicting the total prevalence of ASCs exhibiting a membrane-bound and/or intracellular localization of anti-CD31 IMPs, and the corresponding proportions of each. Values represent mean ± standard deviation of the total prevalence. Triangles depict an internalized localization, and diamonds depict a membrane-bound localization. *$p < 0.05$ compared with suspension. **d** Incorporation of thymidine analogue 5-ethynyl-2'-deoxyuridine (EdU) during deoxyribonucleic acid (DNA) synthesis by IMP-laden and IMP-free ASCs after 48 hr in culture. ns represents not statistically significant. All values represent mean ± standard deviation. All experiments were performed in biological triplicate, using cells derived from three different donors ($n = 3$ biologically independent samples).

antibodies were conjugated to superparamagnetic microparticles through a DNA linker, generating cleavable (c)IMPs. These anti-CD93 cIMPs were slightly but significantly larger in modal diameter than the commercial anti-CD31 IMPs ($4.8 \pm 0.5$ µm vs. $4.4 \pm 0.6$ µm, respectively; $p < 0.0001$), prompting the generation of anti-CD31 cIMPs of a comparable size ($4.8 \pm 0.6$ µm; $p = 0.3772$ compared with anti-CD93 cIMPs; Fig. 7g). The binding and internalization of anti-CD93 cIMPs by ASCs were not significantly different from that of the anti-CD31 cIMPs after 20 min in suspension nor 48 hr in culture, although the internalization of both (anti-CD31 and anti-CD93) cIMPs was significantly lower than that of the smaller anti-CD31 IMPs after

48 hr in culture (Fig. 7h). These findings suggest that the binding and internalization of IMPs by ASCs is non-specific, but is dependent on the size of the microparticles and their exposure to the cells.

The effect of IMP size on their uptake by ASCs was further explored (Fig. 8; Supplementary Fig. 9). Five distinct anti-CD31 IMPs ranging in modal diameter from $0.9 \pm 0.3$ µm to $8.7 \pm 1.5$ µm were investigated (Fig. 8a; Supplementary Table 2). There was an inverse relationship between IMP size and their uptake by ASCs (Fig. 8b; $p < 0.0001$). Interestingly, IMP size also had a significant effect on the temporal dynamics underlying their uptake (Fig. 8b; $p = 0.0003$). While the uptake of anti-CD31

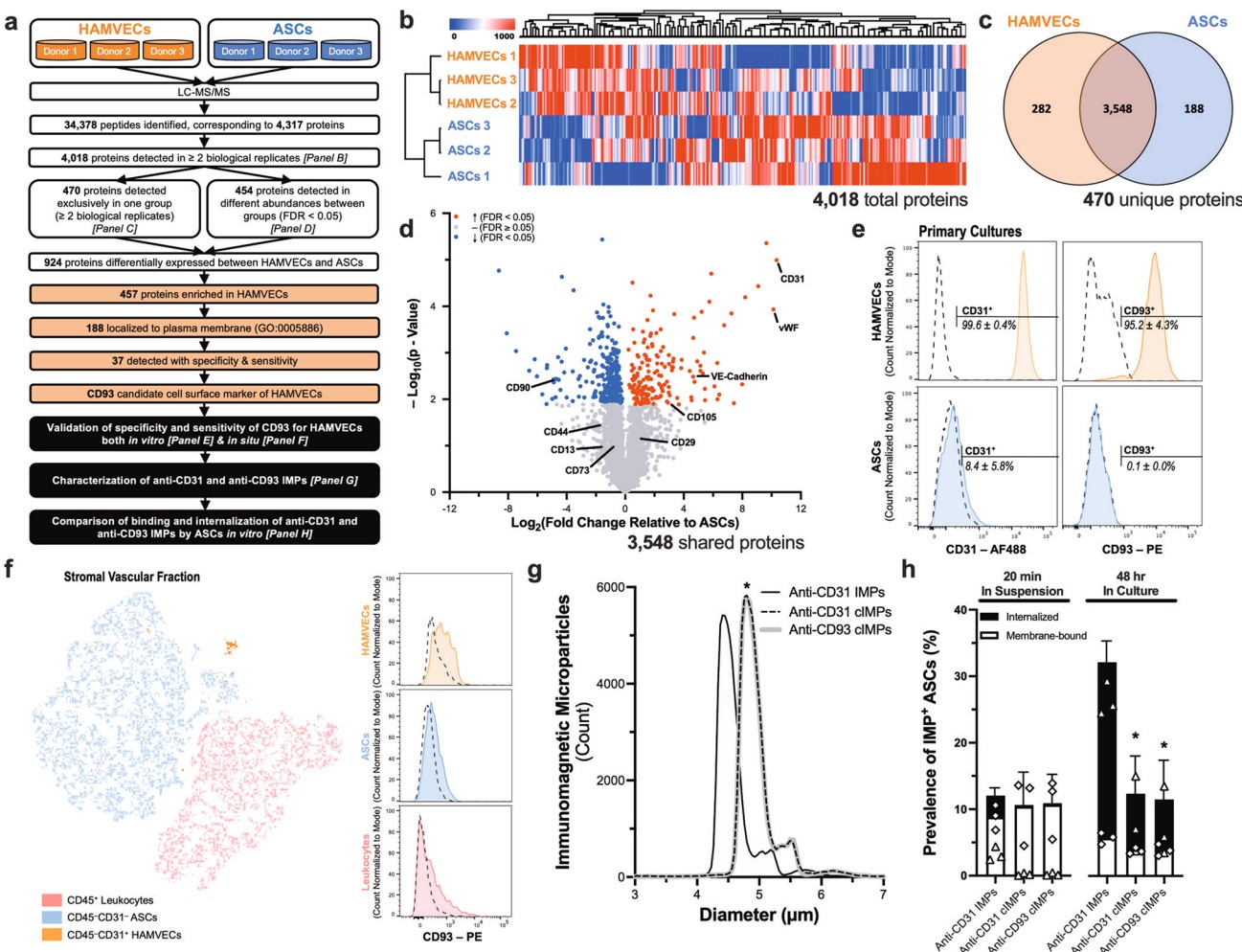

**Fig. 7 Uptake of immunomagnetic microparticles (IMPs) by adipose tissue-derived stromal/stem cells (ASCs) is non-specific, but size and exposure dependent. a** Workflow depicting the proteomic comparison of human adipose tissue-derived microvascular endothelial cells (HAMVECs) and ASCs used to identify immunophenotypic markers with specificity for HAMVECs, which were then validated both in vitro and in situ before generating IMPs and evaluating their binding and internalization by ASCs. **b** Unsupervised hierarchical clustering of the global proteomes of HAMVECs and ASCs. **c** Distribution of proteins detected in HAMVECs and ASCs. **d** Relative abundances of proteins detected in both HAMVECs and ASCs. **e** Cell-surface expression of CD31 and CD93 by HAMVECs and ASCs in culture. Solid lines represent stained cells; dashed lines, isotype controls. **f** Specificity and sensitivity of CD93 for HAMVECs in the stromal vascular fraction of enzymatically digested human subcutaneous abdominal white adipose tissue. Presented is a $t$-distributed stochastic neighbour embedding (tSNE) plot depicting the unsupervised clustering of stromal vascular cells based on their cell-surface expression of CD45, CD31, and CD93, as well as histograms depicting the cell-surface expression of CD93 amongst the three principal subpopulations: CD45−CD31+ HAMVECs, CD45−CD31− ASCs, and CD45+ leucocytes. Solid lines represent stained cells; and dashed lines, fluorescence minus one controls. **g** Size distributions of IMPs. Commercial anti-CD31 IMPs were compared with cleavable (c)IMPs generated to be reactive against CD31 or CD93. *$p < 0.05$ compared with anti-CD31 IMPs. **h** Binding and internalization of IMPs by ASCs after 20 min in suspension (i.e. labelling conditions for magnet-assisted cell sorting procedure) and 48 hr in culture. Shown is a stacked bar graph depicting the total prevalence of ASCs exhibiting a membrane-bound and/or intracellular localization of the IMPs, and the corresponding proportions of each. Values represent mean ± standard deviation of the total prevalence. Triangles depict an internalized localization, and diamonds depict a membrane-bound localization.*$p < 0.05$ compared with anti-CD31 IMPs. **e, h** Values represent mean ± standard deviation. All experiments were performed in biological triplicate, using cells derived from three different donors ($n = 3$ biologically independent samples).

IMPs ≥ 4.4 μm in diameter was exacerbated in culture, those that were ≤3.9 μm in diameter were readily taken up by ASCs after 20 min in suspension, and their culture did not significantly increase their uptake (Fig. 8b). This may be attributed to a greater capacity for ASCs to bind and retain smaller IMPs at the plasma membrane prior to their internalization (Supplementary Fig. 9e). These findings suggest that IMPs ≤ 3.9 μm in diameter cannot be used to isolate HAMVECs due to the capacity for ASCs to readily bind them during the labelling portion of the MACS procedure (i.e. 20 min in suspension), and that the non-specific uptake of

anti-CD31 IMPs ≥ 4.4 μm in diameter can be mitigated by either limiting their exposure to the cells to 20 min in suspension or by using superparamagnetic microparticles of a larger diameter.

**Mitigating the non-specific uptake of IMPs enabled the enrichment of HAMVECs.** The acquisition of HAMVECs by MACS is a two-step procedure: the first step involves the isolation of HAMVECs from the stromal vascular fraction of enzymatically digested fat in order to establish their primary cultures, and the second step involves the enrichment of the primary

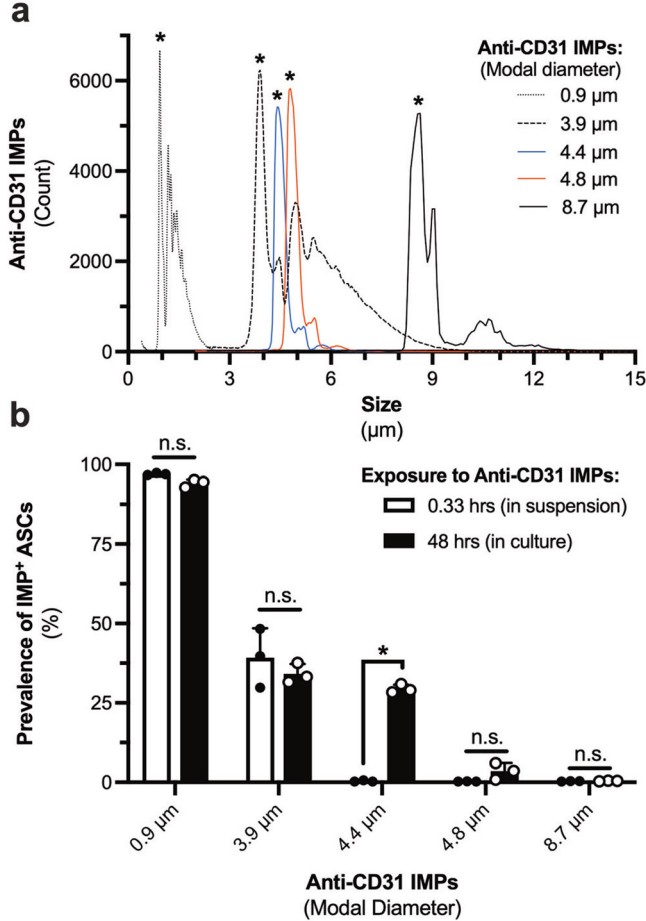

**Fig. 8 Size affects the extent and temporal dynamics underlying the uptake of immunomagnetic microparticles (IMPs) by adipose tissue-derived stromal/stem cells (ASCs). a** Size distribution of five distinct anti-CD31 IMPs. *$p < 0.05$ compared with all other IMPs. **b** Total uptake (i.e. membrane-bound and/or internalized) of the different anti-CD31 IMPs by ASCs after 20 min in suspension (i.e. labelling conditions for magnet-assisted cell sorting) and 48 hr in culture. Values represent mean ± standard deviation; *$p < 0.05$; and, ns represents not significant. This experiment was performed in biological triplicate, using cells derived from three different donors ($n = 3$ biologically independent samples).

cultures in order to eliminate any residual, contaminating ASCs. The limited endothelial plasticity of ASCs and their capacity to bind and internalize IMPs were hypothesized to undermine the enrichment of contaminated cultures of HAMVECs. Our preceding experiments suggest that the uptake of anti-CD31 IMPs by ASCs is non-specific, but size and exposure-dependent. Hence, the effects of alternative target antigens (CD31 vs. CD93), sizes (4.4 μm vs. 4.8 μm), and exposures (suspension vs. culture) of IMPs on the enrichment efficacy of the MACS procedure were investigated using an in vitro model of contaminated primary cultures (Fig. 9a). Specifically, HAMVECs and ASCs were seeded in a 9:1 proportion with or without IMPs to recapitulate their potential exclusion from cultures, made possible by enzymatically cleaving the DNA linkers coupling the antibodies to the superparamagnetic microparticles. Their enrichment was performed after 4 days, and their purities (% CD31+) were evaluated after 7 days (Fig. 9b; Supplementary Fig. 10). In the absence of their enrichment, the purity of the co-cultures deteriorated to 40.0 ± 26.0% (Fig. 9c), underscoring the need for seeding purities >90% for the absolute population growth rate of HAMVECs to exceed that of ASCs.

MACS was effective in increasing the purity of the co-cultures, and there was no significant difference between enriching for HAMVECs on the basis of their expression of CD31 and CD93 (Fig. 9c). Moreover, there was no significant difference between the enrichment of IMP-free cultures and those laden with the larger 4.8 μm (anti-CD31 and anti-CD93) cIMPs, but their enrichment was more effective than that of cultures laden with the smaller 4.4 μm (anti-CD31) IMPs (Fig. 9d). Although not statistically significant, the enrichment of IMP-free cultures yielded higher purities with less variability than the enrichment of 4.8 μm (anti-CD31 and anti-CD93) cIMP-laden cultures (98.9 ± 0.7% vs. 91.0 ± 8.2%, respectively; $p = 0.4612$; Fig. 9d), reflecting the greater capacity of ASCs to internalize 4.8 μm (anti-CD31 and anti-CD93) cIMPs in culture than in suspension (8.1 ± 5.0% after 48 hr in culture vs. 0.2 ± 0.2% after 20 min in suspension; $p = 0.0123$; Fig. 7h). These findings support the non-specific but size and exposure-dependent uptake of IMPs by ASCs, and indicate that the limited capacity of ASCs to express CD31 does not necessitate the utilization of an alternative cell-surface marker with greater specificity for cultured HAMVECs. By mitigating the introduction of ASCs into primary cultures and by enabling their effective sequential enrichment, the DNase-mediated exclusion of anti-CD31 cIMPs from cultures has been used to reliably acquire HAMVECs with high purities from an additional five consecutive patients (98.7 ± 0.5% CD45−CD31+; $n = 5$).

## Discussion

Adipose tissue is an attractive source of ECs for vascular tissue engineering because it can be harvested autologously in large quantities with minimally invasive procedures[10]. While the need for the culture-mediated expansion of HAMVECs is mitigated by the abundant and uniquely dispensable nature of the tissue, the low prevalence of HAMVECs has continued to complicate their acquisition, with their primary cultures often being readily overgrown by fibroblast-like stromal cells[3,11–15]. Here, we demonstrate that the non-specific uptake of IMPs by these residual ASCs from the cell sorting procedure undermines the efficacy of sequential enrichments for HAMVECs. The non-specific uptake of IMPs can be mitigated through the use of superparamagnetic microparticles of a larger diameter or by excluding them from primary cultures where they can be more readily internalized by ASCs. Both of these strategies can mitigate the non-specific uptake of IMPs and can be easily implemented to facilitate the reliable acquisition of HAMVECs in large quantities with high purities for vascular tissue engineering.

MACS was used for the immunoselection of HAMVECs due to its accessibility. Its low cost, small physical footprint, and ease of use make it more amenable for a number of stakeholders when compared with alternatives such as fluorescence-assisted cell sorting. This is an important consideration in promoting the clinical translation of tissue-engineered products and is underscored by the sustained popularity of HUVECs amongst the biomaterials and tissue engineering communities[4,19] in the absence of a readily accessible and clinically viable alternative. The use of IMPs for both the isolation and sequential enrichment of HAMVECs is not only accessible, but may also be more scalable and less variable than other techniques such as differential adhesion[11,15,21], clonal selection[23], and manual weeding[12,22], which are labour-intensive and susceptible to human error in discriminating endothelium from stromal cells.

Immunoselection eliminates human error in the discrimination of cell types by exploiting their cell-surface protein signatures. The enrichment of HAMVECs was pursued rather than the

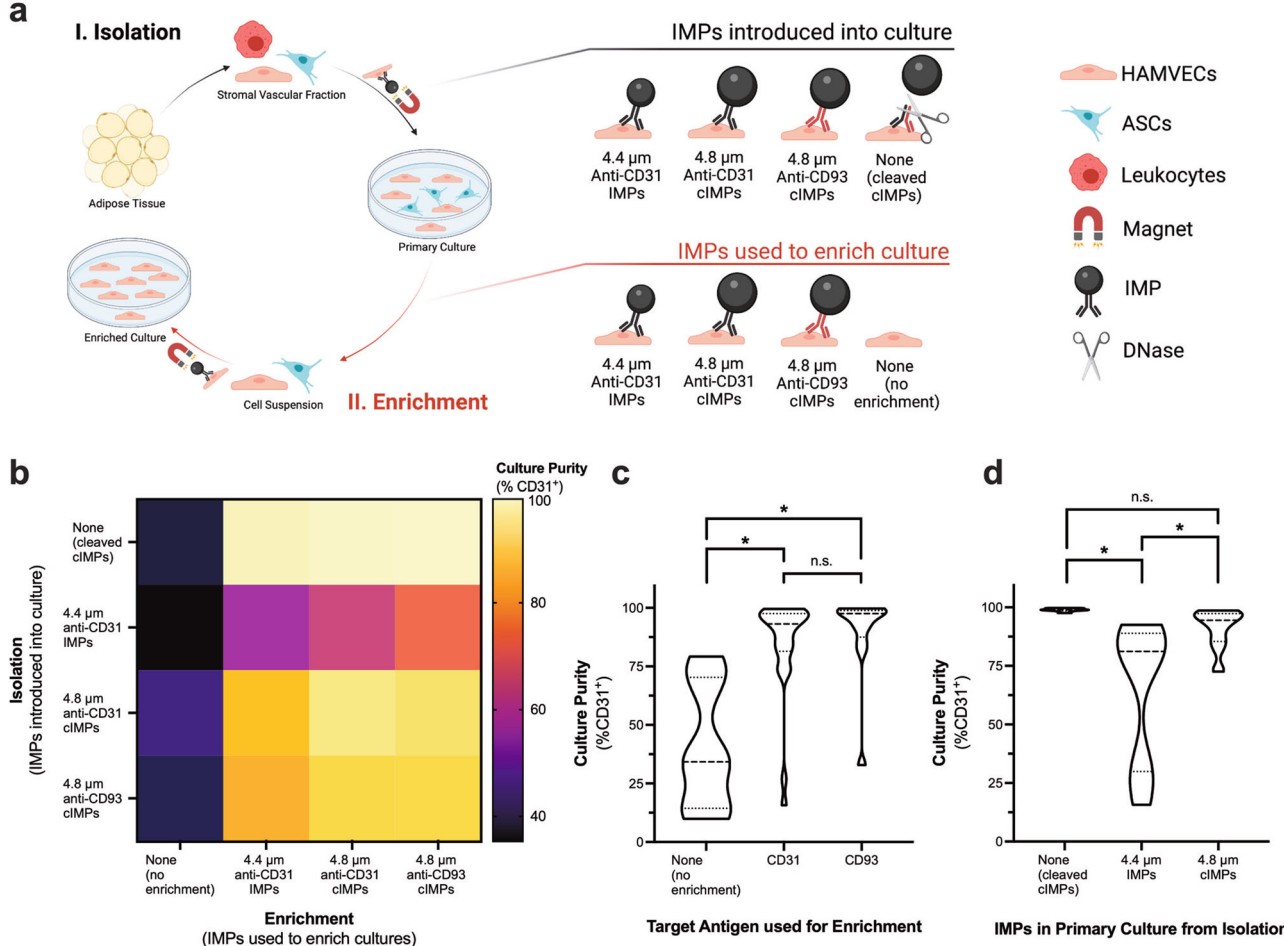

**Fig. 9 Mitigating the non-specific uptake of immunomagnetic microparticles (IMPs) by adipose tissue-derived stromal/stem cells (ASCs) facilitates the enrichment of human adipose tissue-derived microvascular endothelial cells (HAMVECs). a** Schematic depicting the magnet-assisted cell sorting (MACS) procedure used to acquire HAMVECs from enzymatically digested fat, as well as the different combinations of IMPs tested for the isolation and enrichment of HAMVECs using an in vitro model of their contaminated primary cultures. Specifically, HAMVECs and ASCs were seeded in a 9:1 proportion with or without IMPs to recapitulate their potential exclusion from primary cultures, made possible by enzymatically cleaving the deoxyribonucleic acid linkers coupling the antibodies to the superparamagnetic microparticles (i.e. cleavable (c)IMPs). Their enrichment was performed after 4 days, and their purities (% CD31$^+$) were evaluated after a total of 7 days. Created with BioRender.com **b** Effects of alternative target antigens (CD31 vs. CD93), sizes (4.4 μm IMPs vs. 4.8 μm cIMPs), and exposures (introduced vs. excluded from cultures) of IMPs on the enrichment efficacy of the MACS procedure. Values represent mean. **c** Effect of the target antigen on the enrichment efficacy of the MACS procedure. 'None' depicts the distribution of purities of co-cultures that were not subjected to MACS after 4 days; 'CD31', those that were enriched using anti-CD31 IMPs or anti-CD31 cIMPs; and, 'CD93', those that were enriched using anti-CD93 cIMPs. **d** Effect of introducing IMPs into primary cultures on the efficacy of their sequential enrichment. 'None' depicts the distribution of purities of enriched co-cultures that were free of IMPs and cIMPs at the time of MACS; '4.4 μm IMPs', those that were laden with the smaller anti-CD31 IMPs; and, '4.8 μm cIMPs', those that were laden with the larger anti-CD31 cIMPs or anti-CD93 cIMPs. *$p < 0.05$; ns represents not statistically significant. Dashed lines in the violin plots represent median; dotted lines, quartiles; and, horizontal solid lines, range. The experiment was performed in biological triplicate, using cells derived from three different donors ($n = 3$ biologically independent samples).

depletion of non-endothelial cell types due to the heterogeneity of the stromal vascular fraction. Immunophenotyping revealed that HAMVECs account for only 1% of the stromal vascular fraction, with the remainder comprising leucocytes and ASCs. Single-cell RNA sequencing has highlighted that the heterogeneity of the stromal vascular fraction is even greater, with each of these three groups of cells being comprised of multiple subpopulations[37]. The challenge in identifying a cell-surface protein signature that encompasses the phenotypic heterogeneity of non-endothelial cell types may account for the failure of negatively selecting against CD14 and F11 to obtain HAMVECs with purities >90%[13,14].

The purity of HAMVECs is of marked importance to the protracted stability of their cultures. While the faster rate of proliferation of ASCs and their potent growth inhibition of HAMVECs facilitates their overgrowth of primary cultures,

HAMVECs induce a reciprocal, albeit weaker, suppression of ASC proliferation. Therefore, the overgrowth of primary cultures may be prevented by achieving a threshold of purity beyond which the absolute population growth rate of HAMVECs exceeds that of ASCs. Although Arts et al. have suggested that a purity ≥ 75% is sufficient, this conclusion was based on the thicknesses of tissues generated after 3 weeks of co-culturing HUVECs and human foetal lung fibroblasts in several pre-defined proportions[14]. In contrast, our co-cultures of HAMVECs and ASCs exhibited a deterioration in purity from 90 to 40% after only 1 week. This discrepancy may be attributed to the differential rates of proliferation and disparate cell–cell interactions between HUVECs and human foetal lung fibroblasts vs. HAMVECs and ASCs. Our primary cultures of HAMVECs could be repeatedly sub-cultured and maintained at confluence for 4 weeks without

any signs of stromal cell overgrowth if established with purities ≥98%.

The paucity of markers with specificity and sensitivity for the endothelial lineage complicated the acquisition of HAMVECs. The stromal vascular fraction had to be depleted of CD45[+] leucocytes prior to positively selecting for CD31[+] HAMVECs due to their capacity to co-express this characteristic endothelial marker. Furthermore, contaminating ASCs from the cell sorting procedure were found to express characteristic endothelial markers, including CD31, VE-cadherin, and vWF. Although the temporal dynamics underlying the expression of these conventional phenotypic markers were not investigated, the differential detection of CD93 in vitro and in situ supports that the action of culture itself induces a phenotypic change in both ECs[38] and ASCs[39]. However, the extent to which this phenomenon is a manifestation of their phenotypic plasticity versus the culture-mediated selection for their different subpopulations remains unclear.

The phenotypic heterogeneity of the stromal vascular fraction is not limited to ASCs and leucocytes, but is also evident amongst the ECs[37]. CD31 is a pan-endothelial marker[1], suggesting that HAMVECs isolated from the stromal vascular fraction on the basis of a CD45[−]CD31[+] immunophenotype comprise a heterogeneous mixture of arterial, venous, and lymphatic subpopulations. Interestingly, CD93 has been reported to preferentially stain post-capillary venules[36], suggesting that its greater sensitivity for HAMVECs in vitro than in situ may reflect the culture-mediated selection for the venous subpopulation. The phenotypic heterogeneity of CD45[−]CD31[+] HAMVECs is an important area for further research as it may mitigate the phenotypic heterogeneity observed between ECs derived from different vascular beds[1,35]; i.e., the putatively heterogeneous CD45[−]CD31[+] HAMVECs may be further enriched for desirable phenotypic specializations for tailored vascular tissue engineering applications[3].

The capacity of anti-CD31 IMPs to establish primary cultures of HAMVECs but not facilitate their subsequent enrichment was believed to have been the manifestation of the *de novo* expression of CD31 by ASCs induced by their culture in medium containing vascular endothelial growth factor and basic fibroblast growth factor[29]. However, despite detecting CD31 in cultured ASCs by LC–MS/MS and validating its cell-surface localization by flow cytometry, its efficacy as a target antigen for the enrichment of HAMVECs was comparable to that of CD93. The comparable enrichment efficacies of anti-CD31 IMPs and anti-CD93 IMPs despite the greater specificity of CD93 for cultured HAMVECs may be attributed to the low level of expression of CD31 by ASCs. The abundance of CD31 in ASCs was <0.08% of that in HAMVECs, requiring secondary antibody-mediated signal amplification of the directly conjugated anti-CD31 antibody to facilitate its detection by flow cytometry. The binding affinity of antibodies is dependent on the antigen density present on the cell surface[40], suggesting that the cell-surface density of CD31 amongst ASCs may have been too sparse to facilitate their stable binding of the anti-CD31 IMPs.

The sequential enrichment of primary cultures was undermined by the non-specific uptake of the anti-CD31 IMPs. The internalization of particles is dependent on not only their size, shape, and surface chemistry[41], but also the cell type[42]. While the capacity for leucocytes to uptake micron-sized particles is well-established[43], their internalization by other cell types has been largely dismissed[44]. The commercial anti-CD31 IMPs utilized in this investigation were 4.4 μm in diameter (volume: 45 μm$^3$) and were readily internalized by over 25% of ASCs. Subsequent enrichments for CD31 expression consequently selected for not only HAMVECs, but also the ASCs that had internalized the anti-CD31 IMPs. While the non-specific internalization of the anti-CD31 IMPs could be mitigated through the use of microparticles

of a larger diameter (e.g. diameter: 4.8 μm; volume: 58 μm$^3$), their exclusion from cultures, made possible by enzymatically cleaving DNA linkers coupling the antibodies to the superparamagnetic microparticles, was found to be most effective in facilitating the acquisition of HAMVECs with the highest purity and least variability.

The low prevalence of microvascular ECs in tissues has remained a formidable obstacle to their reliable acquisition, prompting many to turn to alternative sources of endothelium for vascular tissue engineering at the expense of increased regulatory scrutiny. The challenges underlying their isolation and expansion were investigated to develop an accessible and reliable method of obtaining them from human fat—an abundant and uniquely dispensable source of autologous endothelium for the vascularization of tissue-engineered constructs and the endothelialization of small-diameter vascular prostheses. Although disparate growth kinetics and the paucity of markers with specificity and sensitivity for the endothelial lineage challenged their acquisition, mitigating the non-specific uptake of IMPs was imperative for the effective sequential enrichment of HAMVECs to purities that prevented their overgrowth by ASCs. The findings of this study demonstrate the feasibility of sourcing autologous endothelium from human fat, and delineate a reliable and facile method for its acquisition from patients that can be readily applied in future vascular tissue engineering applications.

## Methods

**Materials**. Subcutaneous abdominal white adipose tissue was obtained with informed consent from 25 patients undergoing reconstructive breast surgery at the University Health Network (Toronto, ON, Canada; institutional research ethics board approval no. 13-6437-CE). Tissue culture-treated polystyrene (TCPS) was sourced from Corning (Corning, NY, United States). Unless indicated otherwise, all other materials were from Sigma-Aldrich (St. Louis, MO, United States).

**Isolation of the stromal vascular fraction**. The stromal vascular fraction was isolated from adipose tissue as previously described[29]. Briefly, adipose tissue was minced and enzymatically digested for 1 hr at 37 °C using collagenase type II (2 mg/mL) in Kreb's Ringer bicarbonate buffer supplemented with 3 mM glucose, 25 mM 4-(2-hydroxyethyl)-1-piperazineethanesulfonic acid, and 20 mg/mL bovine serum albumin (BSA). The buoyant adipocytes were discarded following the centrifugation of the digest, and the pelleted tissue was subjected to another 15 min of enzymatic digestion at 37 °C using 0.25% trypsin-ethylenediaminetetraacetic acid (EDTA) solution. The cells were then resuspended in sterile distilled and deionized water supplemented with 0.154 M ammonium chloride, 10 mM potassium bicarbonate, and 0.1 mM EDTA for 10 min to facilitate erythrocyte lysis, before being filtered through a 100 μm sieve. The resulting filtrate was defined as the stromal vascular fraction and was immediately prepared for MACS or immunophenotyping.

**Magnet-assisted cell sorting**. IMPs directed against CD45 (Dynabeads 45) and CD31 (Dynabeads CD31 Endothelial Cell) were obtained from Invitrogen (Carlsbad, CA, United States). Alternatively, cIMPs were generated using the CELLection Biotin Binder Kit (Invitrogen). Specifically, 5 μg of either mouse anti-human CD31 antibodies (Miltenyi Biotec, Bergisch Gladbach, Germany; catalogue no. 130-119-893) or mouse anti-human CD93 antibodies (BioLegend, San Diego, CA, United States; catalogue no. 336104) were conjugated to 10$^7$ Dynabeads via biotin-streptavidin interactions through DNA linkers, enabling their release from cells with the supplied DNase.

HAMVECs and ASCs were isolated from the stromal vascular fraction using MACS. Cells were resuspended in sorting buffer (phosphate-buffered saline without calcium chloride and magnesium chloride (PBS$^{−/−}$) supplemented with 2 mM EDTA and 0.1% (w/v) BSA), and were incubated with IMPs for 20 min at 4 °C before being magnetically separated using the DynaMag-5 Magnet (Invitrogen). The stromal vascular fraction was depleted of CD45[+] leucocytes prior to separating CD45[−]CD31[+] HAMVECs from CD45[−]CD31[−] ASCs. The enzymatic exclusion of cIMPs from cultures was performed in select experiments using the supplied DNase as per the manufacturer's instructions. Briefly, cIMP-labelled cells were incubated with DNase to dissociate the cells from the superparamagnetic microparticles prior to using the DynaMag-5 Magnet to separate the superparamagnetic microparticles from the cells of interest. The latter were then seeded onto TCPS in the absence of the superparamagnetic microparticles used to isolate them. Cultures of HAMVECs were enriched using IMPs or cIMPs directed against CD31 or CD93 using the same procedure.

**Cell culture**. HAMVECs and ASCs were plated onto TCPS and cultured at 37 °C, 5% $CO_2$ under a relative humidity of 85% in Endothelial Cell Growth Medium (EGM)-2 (Lonza, Walkersville, MD, United States). HUVECs (Lonza; $n = 3$ biologically independent samples), HCAECs (Lonza; $n = 3$ biologically independent samples), and HDMVECs (Lonza and PromoCell, Heidelberg, Baden-Württemberg, Germany; $n = 3$ biologically independent samples) were obtained commercially and cultured under the same conditions. Media was exchanged three times a week, and cells were passaged at 75–90% confluence using TrypLE Express (Invitrogen). Phase-contrast transmission light microscopy was used to assess morphology and confluence (Leica DMIL, Leica Microsystems, Wetzlar, Hesse, Germany). Cells were counted using a hemocytometer, and dead cells were excluded based on trypan blue staining.

**Immunophenotyping**. Cells were stained with either Live/Dead Fixable Aqua or Live/Dead Fixable Near-IR (Invitrogen), and Fc receptors were blocked using Human TruStain FcX (BioLegend). Cells were then stained for 20 min at 4 °C with combinations of the following fluorophore-conjugated mouse anti-human monoclonal antibodies sourced from BioLegend: CD45-APC/Cy7 (catalogue no. 368516), CD45-Brilliant Violet 785 (catalogue no. 304048), CD31-Alexa Fluor 488 (catalogue no. 303110), CD31-Brilliant Violet 421 (catalogue no. 303124), and CD93-PE (catalogue no. 336108). The staining concentration for each of the antibodies was 5 µg/mL. For select experiments, the signal of the Alexa Fluor 488-conjugated mouse anti-human CD31 antibody (BioLegend; catalogue no. 303110; staining concentration, 5 µg/mL) was amplified using an Alexa Fluor 488-conjugated goat anti-mouse IgG secondary antibody (Invitrogen; catalogue no. A11001; staining concentration, 5 µg/mL; Supplementary Fig. 8). Stained cells were fixed with 4% (w/v) paraformaldehyde in PBS$^{-/-}$ for 15 min at 4 °C. Compensation was achieved using the AbC Anti-Mouse Bead Kit and the ArC Amine Reactive Compensation Bead Kit (Invitrogen). Gates were set using fluorescence minus one controls. Flow cytometry was performed using a BD LSR II or BD LSR Fortessa flow cytometer (Becton, Dickinson and Company, Franklin Lakes, NJ, United States) at the Temerty Faculty of Medicine Flow Cytometry Facility (University of Toronto, Toronto, ON, Canada). Data was acquired using BD FACSDiva software version 8.0.1, and analysed using FlowJo software version 10.7.1 (Becton, Dickinson and Company). The immunophenotypes of two populations of cells were compared using two-tailed $t$-tests, blocking for donors when appropriate; and, those of ≥ 3 populations were compared using a one-way analysis of variance (ANOVA), with Tukey's post-hoc test for multiple comparisons.

**Gene expression**. Reverse transcription quantitative real-time polymerase chain reaction was performed as previously described[29]. Briefly, total ribonucleic acid isolated from cells using Trizol (Invitrogen) was immediately reverse transcribed into complementary DNA (cDNA) using the High Capacity cDNA Reverse Transcription Kit (Applied Biosystems, Foster City, CA, United States). The quantitative real-time polymerase chain reaction was performed on a CFX384 Touch Real-Time PCR Detection System (Bio-Rad, Hercules, CA, United States), using the SsoAdvanced Universal SYBR Green Supermix (Bio-Rad). Each reaction comprised 10 ng template cDNA and 450 nM of both forward and reverse primers in a total volume of 10 µL, with thermal cycling performed as previously described[29]. The primers employed were previously designed and validated to amplify GAPDH, PECAM1, CDH5, and VWF[29]. Their nucleotide sequences are delineated in Supplementary Table 1. All experiments utilized three technical replicates for each biological sample and included no reverse transcriptase controls. Data were analysed using Bio-Rad CFX Maestro 1.1 software version 4.1.2433.1219 (Bio-Rad).

The abundance of messenger ribonucleic acid encoding genes of interest in the putative HAMVECs was compared to that in HUVECs, HCAECs, and HDMVECs using a two one-sided test for equivalence[45]. Specifically, equivalence was established within the significance level $\alpha$ when the $(1-2\alpha) \times 100\%$ confidence interval of the difference in mean quantification cycles ($C_q$) between HAMVECs and the EC controls was contained within the equivalence margin ($\pm \partial$). The significance level $\alpha$ was set to 0.05, and the equivalence margin $\partial$ was set to three standard deviations of the Gaussian distribution of $C_q$ values amongst the EC controls about their gene-normalized mean (Supplementary Fig. 3). The EC controls were evaluated separately but statistically presented as a single population in order to ascertain the variability of their expression of characteristic endothelial genes[29]. GAPDH was used as a loading control.

**Immunofluorescence**. Cells were rinsed with phosphate-buffered saline (with calcium chloride and magnesium chloride; PBS$^{+/+}$), fixed with ice-cold methanol for 10 min at − 20 °C, and blocked with 3% (w/v) BSA in PBS$^{-/-}$ for 30 min prior to being stained overnight at 4 °C with 5 µg/mL of primary antibodies in the same blocking solution. The primary antibodies were sourced from Abcam, and included a mouse anti-human CD31 antibody (catalogue no. ab24590), a rabbit anti-human VE-cadherin antibody (catalogue no. ab33168), and a mouse anti-human vWF antibody (catalogue no. ab194405). Cells were then blocked with Normal Serum Block (BioLegend) for 30 min at 25 °C and stained for 1 hr with 2.5 µg/mL of either a goat anti-mouse IgG-Alexa Fluor 594 secondary antibody (BioLegend; catalogue no. 405326) or a goat anti-rabbit IgG-Alexa Fluor 488 secondary antibody (Abcam;

catalogue no. ab150077) diluted in 3% (w/v) BSA in PBS$^{-/-}$. Cells were then stained for 5 min with 3 µM 4′,6-diamidino-2-phenylindole (Abcam) in PBS$^{-/-}$. Wide-field immunofluorescence microscopy was performed using a Leica DMi8, operated using the Leica Application Suite X software version 3.5.5.19976 (Leica Microsystems). Image processing was performed using Fiji software version 2.1.0/1.53c[46].

**Acetylated low-density lipoprotein uptake**. AcLDL uptake was assessed as previously described[29]. Cells were incubated with 10 µg/mL Alexa Fluor 488-conjugated AcLDL (Invitrogen) in EGM2 for 4 hr, after which they were fixed with 4% (w/v) paraformaldehyde in PBS$^{-/-}$ for 15 min at 4 °C. Cells were analysed by flow cytometry.

**Angiogenic capacity**. The angiogenic capacity of cells was assessed as previously described[47]. Briefly, TCPS was coated with 150 µL/cm² of Cultrex PathClear Basement Membrane Extract (Bio-Techne, Minneapolis, MN, United States). Cells were then seeded at a density of 45,000 cells/cm² and incubated in EGM2 for 6 hr before their formation of capillary-like tubules was assessed using phase-contrast transmission light microscopy (Leica DMIL).

**Adipogenic plasticity**. The accumulation of lipids by ASCs and HAMVECs was assessed using a protocol adapted from Kraus et al.[48]. Cells were seeded onto TCPS at a density of 4,000 cells/cm² and cultured in EGM2 for 24 hr before the media was replaced with StemPro Adipogenesis Differentiation Kit (Invitrogen). The latter was exchanged every other day for 10 days before cells were rinsed with PBS$^{-/-}$ and fixed with 4% (w/v) paraformaldehyde in PBS$^{-/-}$ for 15 min at room temperature. Fixed cells were rinsed with 40% (v/v) isopropanol in distilled water before being stained with 0.2% (w/v) Oil Red O in 40% (v/v) isopropanol in distilled water for 30 min at room temperature. Stained cells were rinsed with distilled water before being counterstained with Mayer's hematoxylin (Abcam; Cambridge, United Kingdom) for 5 min at room temperature. Cells were rinsed with distilled water, and representative photomicrographs were acquired using brightfield transmission light microscopy (Leica DMIL). Oil Red O was then eluted from the cells by incubating them in 100% isopropanol for 10 min at room temperature, and the absorbance of the eluent at 515 nm was quantified by spectrophotometry (EnVision 2104 Multilabel Reader operated using EnVision Manager software version 1.14.3049.528; PerkinElmer, Waltham, MA, United States). The absorbance of the eluent was compared using a two-way ANOVA with Tukey's post-hoc test for multiple comparisons, blocking for donors.

**Liquid chromatography–tandem mass spectrometry**. Reversed-phase LC–MS/MS was performed as previously described[29]. Briefly, cells were lysed and proteins extracted in 50% (v/v) 2,2,2-trifluoroethanol in PBS$^{-/-}$ supplemented with 100 mM ammonium bicarbonate. Proteins were reduced with 5 mM dithiothreitol, alkylated with 15 mM iodoacetamide, and digested with mass spectrometry-grade Trypsin-Lys C mix (Promega, Madison, WI, United States) following their dilution with 100 mM ammonium bicarbonate and supplementation with 2 mM calcium chloride. Formic acid was used to quench the digestion, after which the tryptic peptides were de-salted using OMIX C18 solid-phase extraction tips (Agilent, Santa Clara, CA, United States). Samples were dried by vacuum centrifugation and reconstituted in 5% (v/v) formic acid in high-performance liquid chromatography-grade water.

Tryptic peptides were analysed on an Easy-nLC 1200 (Thermo Fisher Scientific, Waltham, MA, United States) coupled to a Q Exactive Plus mass spectrometer (Thermo Fisher Scientific) through a Nanospray Flex Ion Source (Thermo Fisher Scientific). Tryptic peptides were loaded onto an in-house packed reversed-phase 10-cm, 75 µm internal diameter column (Reprosil-Pur Basic C18, 3 µm, 100 Å; Dr. Maisch HPLC, Ammerbuch, Baden-Württemberg, Germany), and separated using a 3-hr acetonitrile linear gradient (2–35% (v/v) in 0.1% (v/v) formic acid in high-performance liquid chromatography-grade water) at 250 nL/min. All experiments utilized two technical replicates for each biological sample. Spectra were collected using a top ten data-dependent acquisition method as previously described[29], using Tune software version 2.8 (Thermo Fisher Scientific) and Xcalibur software version 4.0.27.19 (Thermo Fisher Scientific). Raw files were searched against the UniProt human proteome database (updated to 2017-07-24) using MaxQuant software version 1.6.0.1 (Max Planck Institute of Biochemistry, Planegg, Bavaria, Germany)[49], with 'match between runs' enabled. Cysteine carbamidomethylation was set as a fixed modification, and methionine oxidation, N-terminal acetylation, and asparagine or glutamine deamidation were selected as variable modifications. The false discovery rate (FDR) was set to 1% using a reversed-target decoy database.

Data visualization and statistical analyses were completed using the Perseus 1.6.1.2 software package (Max Planck Institute of Biochemistry)[50]. Label-free quantification values were log$_2$-transformed[51], and missing values were imputed from a normal distribution using a downshift of 1.8 and width of 0.3 standard deviations (Supplementary Data 1 and 2). Unsupervised hierarchical clustering and associated heat maps of proteins that were identified in ≥2 biological replicates in at least one group were generated from normalized values across all samples for each protein. Differentially expressed proteins between groups of cells were defined as

being either uniquely detected (i.e. detected vs. not detected in ≥2 biological replicates per group) or quantified in statistically different abundances (i.e. detected in ≥2 biological replicates in each group, but $p < 0.05$ or FDR < 0.05). Statistically overrepresented ($p < 0.05$) biological pathways (Reactome version 65; released 2019-12-22) among the differentially expressed proteins were identified using the Fisher's Exact test with Bonferroni's correction for multiple testing within PANTHER version 15.0 released 2020-02-14[52], and the Gene Ontological Term Mapper was used to map differentially expressed proteins to their cellular components.

**Population doubling time**. Cells were seeded at a density of 4,000 cells/cm² into seven TCPS flasks, and every 24 hr cells from one flask were counted using a hemocytometer. Population doubling times were calculated from the exponential growth phase using the formula PDT = $[\Delta t \times \log_{10}(2)] \div [\log_{10}(n_f \div n_i)]$, where PDT represents population doubling time, $\Delta t$ represents the duration of exponential growth, and $n_i$ and $n_f$ represent the number of cells at the initiation and termination of the exponential growth phase, respectively[53]. The population doubling times of two populations of cells were compared using two-tailed $t$-tests, blocking for donors where appropriate, and those of ≥3 populations were compared using a one-way ANOVA, with Tukey's post-hoc test for multiple comparisons.

**Population growth rates in co-culture**. The effect of seeding purity on the population growth rates of HAMVECs and ASCs was assessed using an assay adapted from Gerashchenko (Supplementary Fig. 4)[54]. HAMVECs and ASCs were seeded onto TCPS in several pre-defined proportions amounting to a total density of 4,000 cells/cm² or separately at their respective constituent densities. Cells were cultured for 4 days, after which the co-cultured cells were stained with 4 μM Cell-Trace Far Red (Invitrogen) and combined with the mono-cultured controls. The mixture was then blocked with Human TruStain FcX (BioLegend), stained with 5 μg/mL of mouse anti-human CD31-Alexa Fluor 488 (BioLegend; catalogue no. 303110) for 20 min at 4 °C, and fixed with 4% (w/v) paraformaldehyde in PBS$^{-/-}$ for 15 min at 4 °C. Cells were analysed by flow cytometry. Population growth rates in co-culture were calculated using the formulas, $R_1 = [\text{Dye}^+\text{CD31}^+ \div \text{Dye}^-\text{CD31}^+] \times [\text{Dye} \div \text{Dye}^-] \times 100$ and $R_2 = [\text{Dye}^+\text{CD31}^- \div \text{Dye}^-\text{CD31}^-] \times [\text{Dye}^+ \div \text{Dye}^-] \times 100$, where $R_1$ and $R_2$ represent the population growth rates of HAMVECs and ASCs relative to their mono-cultured controls, respectively. Two-tailed one-sample $t$-tests were used to compare the population growth rates in co-culture to that in mono-culture (i.e. 100%), and a two-way ANOVA with Bonferroni's post hoc test for multiple comparisons was used to evaluate the relationship between the seeding purity and the resultant population growth rates, blocking for donors.

**Temporal stability of population purities**. The effect of seeding purity on the temporal composition of cultures was assessed. HAMVECs and ASCs were seeded onto TCPS in pre-defined proportions (100 or 90% HAMVECs) amounting to a total density of 4,000 cells/cm². The purity (% CD31$^+$) of the cultures was assessed by flow cytometry after 1, 7, 14, and 28 days. The effect of seeding purity on the purity of the cultures over time was assessed using a two-way ANOVA with Tukey's post-hoc test for multiple comparisons, blocking for donors.

**Immunomagnetic microparticle size distribution**. The size distribution of IMPs was evaluated with a Multisizer 4e Coulter Counter (Beckman Coulter Life Sciences, Indianapolis, IL, United States), using a 100 μm aperture calibrated with 10 μm beads (Coulter CC Size Standard L10; Beckman Coulter Life Sciences) or a 20 μm aperture calibrated with 2 μm beads (Coulter CC Size Standard L2; Beckman Coulter Life Sciences)[55]. Data was acquired for a modal count of 5,000 using the Multisizer 4e software version 4.03 (Beckman Coulter Life Sciences). The polydispersity index (PDI) of the IMPs was calculated using the formula PDI = (standard deviation/mean)², as previously described[55]. The size distributions of the different IMPs were compared using a one-way ANOVA with Tukey's post-hoc test for multiple comparisons.

**Immunomagnetic microparticle uptake and localization**. IMPs were detected on the basis of their autofluorescence, and in select experiments, a membrane-impermeable secondary antibody directed against their binding moiety was used to discriminate between their extracellular and intracellular localization (Supplementary Fig. 5). The autofluorescence of the IMPs and the effect of conjugating the goat anti-mouse IgG-Alexa Fluor 647 secondary antibody (Cell Signalling Technology, Danvers, MA, United States; catalogue no. 4410S; staining concentration, 5 μg/mL) on their fluorescence was characterized by spectral scanning confocal microscopy (Leica SP8 microscope operated by Leica Application Suite X software version 3.5.5.19976, Leica Microsystems; Advanced Optical Microscopy Facility, University Health Network), with 20 nm detection bandwidths employed to assess the emission spectra for each of the 405 nm, 488 nm, 552 nm, and 638 nm excitation wavelengths. 30 IMPs were selected as regions of interest in generating representative emission spectra. Candidate excitation-emission features of the IMPs were validated by flow cytometry.

The localization of IMPs in culture was assessed by confocal microscopy. ASCs were seeded at a concentration of 4,000 cells/cm² onto 35 mm μ-Dishes (ibidi

GmbH, Gräfelfing, Germany) with 400,000 anti-CD31 IMPs/cm² (Dynabeads; Invitrogen). After 24 hr, cells were rinsed with PBS$^{+/+}$ to remove non-adherent IMPs, fixed with 4% (w/v) paraformaldehyde in PBS$^{-/-}$ for 15 min at 4 °C, and permeabilized with 0.1% (v/v) Triton X-100 in PBS$^{-/-}$ for 15 min. F-actin was stained with Alexa Fluor Plus 647 Phalloidin (Invitrogen), and nuclei were stained with 3 μM 4′,6-diamidino-2-phenylindole in PBS$^{-/-}$ for 5 min (Abcam). Anti-CD31 IMPs were detected on the basis of their autofluorescence, and their localization was assessed by z-stacking confocal microscopy. Images were processed using Fiji software version 2.1.0/1.53c[46].

The prevalence of IMP uptake and their localization in ASCs was assessed by flow cytometry (Supplementary Fig. 5). Specifically, $10^5$ ASCs were exposed to $10^7$ IMPs in suspension for 20 min at 4 °C, or in culture in 25 cm² TCPS flasks for pre-defined durations. Cells were then stained with 5 μg/mL of goat anti-mouse IgG-Alexa Fluor 647 secondary antibody (Cell Signalling Technology; catalogue no. 4410S) for 20 min at 4 °C, and fixed with 4% (w/v) paraformaldehyde in PBS$^{-/-}$ for 15 min at 4 °C. Cells were analysed by flow cytometry. The prevalence of IMP localization over time was compared by a two-way ANOVA with Bonferroni's post-hoc test for multiple comparisons, blocking for donors; and, the total prevalence of IMP uptake by ASCs after 20 min in suspension vs. 48 hr in culture was compared using a two-tailed $t$-test, blocking for donors.

**DNA synthesis**. The effect of binding and internalizing IMPs on the proliferative capacity of ASCs was assessed using the Click-iT EdU Alexa Fluor 647 Flow Cytometry Assay Kit (Invitrogen; Supplementary Fig. 6). ASCs were seeded at a concentration of 4,000 cells/cm² onto TCPS with 400,000 anti-CD31 IMPs/cm² (Dynabeads; Invitrogen). After 48 hr in culture, cells were rinsed with PBS$^{+/+}$ to remove free IMPs and pulsed with 10 μM 5-ethynyl-2′-deoxyuridine (EdU) in EGM2 for 6 hr. Cells were then resuspended in cell sorting buffer (2 mM EDTA and 0.1% (w/v) BSA in PBS$^{-/-}$), and IMP-laden ASCs were magnetically separated from IMP-free ASCs. Fixation, permeabilization, and staining were performed using the kit. Cells were analysed by flow cytometry. A two-tailed $t$-test was used to compare EdU incorporation between IMP-laden and IMP-free ASCs, blocking for donors.

**Effect of size on immunomagnetic microparticle uptake**. The effect of IMP size on their uptake by ASCs was assessed. Superparamagnetic microparticles ranging in diameter from 1 to 8 μm (Spherotech, Lake Forest, IL, United States; catalogue no. SVM-10-10, SVM-40-10, and SVM-80-5) were conjugated to mouse anti-human CD31 antibodies (Miltenyi Biotec; catalogue no. 130-119-893) via biotin-streptavidin interactions (5 μg of antibody per $10^7$ particles). Their uptake by ASCs was compared to that of the commercial anti-CD31 IMPs (Dynabeads CD31 Endothelial Cell; Invitrogen) and the anti-CD31 cIMPs. The size distributions of these five distinct anti-CD31 IMPs was assessed using a Coulter counter.

The binding and internalization of the anti-CD31 IMPs was assessed by flow cytometry (Supplementary Fig. 9b and 9c). Specifically, $10^5$ ASCs were exposed to $10^7$ IMPs in suspension for 20 min at 4 °C, or in culture in 25 cm² TCPS flasks for pre-defined durations. Cells were then resuspended in cell sorting buffer (2 mM EDTA and 0.1% (w/v) BSA in PBS$^{-/-}$), and IMP-laden ASCs were magnetically separated from IMP-free ASCs. IMP$^+$ ASCs were stained with 4 μM CellTrace Far Red (Invitrogen) before being re-combined with the IMP$^-$ ASCs. The mixture was then stained with 5 μg/mL of goat anti-mouse IgG-Alexa Fluor 488 secondary antibody (Invitrogen; catalogue no. A11001) for 20 min at 4 °C, and fixed with 4% (w/v) paraformaldehyde in PBS$^{-/-}$ for 15 min at 4 °C. Cells were analysed by flow cytometry. The effect of IMP size on their uptake by ASCs over time was assessed using a two-way ANOVA with Tukey's post-hoc test for multiple comparisons, blocking for donors.

**Enrichment efficacies of alternative MACS strategies**. The effects of alternative target antigens, sizes, and exposures of IMPs on the enrichment efficacy of the MACS procedure were investigated using an in vitro model of contaminated primary cultures. HAMVECs and ASCs were seeded onto TCPS at a density of 4,000 cells/cm² in a 9:1 proportion with or without 400,000 IMPs/cm² to recapitulate their potential exclusion from primary cultures. Their enrichment was performed using MACS after 4 days, and their purities (% CD31$^+$) were evaluated after a total of 7 days by flow cytometry. A two-way ANOVA with Tukey's post-hoc test for multiple comparisons was used to compare the culture purities of the different MACS strategies, blocking for donors. The effects of target antigen and IMP size/exposure were assessed using a one-way ANOVA with Tukey's post-hoc test for multiple comparisons; pooling replicates where applicable.

**Statistics and reproducibility**. Experiments were performed three times using cells derived from three different donors ($n = 3$ biologically independent samples), unless otherwise stated. Other than the proteomics data that was analysed using the Perseus 1.6.1.2 software package (Max Planck Institute of Biochemistry)[50], statistical analyses were performed using Prism 8 software version 8.4.3 (GraphPad Software, San Diego, CA, United States). Where applicable, normality of data was assessed using quantile-quantile plots and the Shapiro–Wilk test. Unless stated otherwise, $p < 0.05$ was accepted as statistically significant and values are represented as mean ± standard deviation.

**Reporting summary**. Further information on research design is available in the Nature Research Reporting Summary linked to this article.

## Data availability

Raw data and search results from the LC–MS/MS are available from the MassIVE repository (accession no. MSV000086982), and the corresponding tabulated datasets are supplied in Supplementary Data 1 and 2. Source data underlying the graphs and charts presented in the main figures are provided in Supplementary Data 3. All other data are available from the corresponding author upon reasonable request.

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

## Acknowledgements

The authors thank Kate Butler for her assistance in facilitating the provision of tissue for this research. This work was funded by the Canadian Institutes for Health Research (grant no. 230762 & 426275). J.A.A. was supported by a Post Graduate Scholarship—Doctoral (PGS-D3) from the Natural Sciences and Engineering Research Council of Canada (NSERC), an Ontario Graduate Scholarship, and an Education Fund award from the Ted Rogers Centre for Heart Research (TRCHR). V.M. was supported by an Undergraduate Student Research Award from NSERC and a Ted Rogers Scholar Award from TRCHR.

## Author contributions

J.A.A., C.A.S. and J.P.S. conceived the study. J.A.A. designed the study. J.A.A., V.M. and M.J.M. collected and analysed data. A.O.G., S.O.P.H. and J.P.S. provided material for the study. J.A.A. wrote the article. All authors reviewed and approved the article.

## Competing interests

The authors declare no competing interests.
