## [Transparent Peer Review File · Communications Biology]

Reviewers' comments:

Reviewer #1 (Remarks to the Author):

Jeremy A. Antonyshyn and co-workers realized a very clear and helpful work for the biomedical and tissues engineering community. They worked on the isolation and enrichment of human adipose tissue-derived microvascular endothelial cells (HAMVECs), rare endothelium cells (almost 1% of the total stromal vascular fraction) for vascular tissue engineering purposes. The challenge addressed here is depicting a finely controlled protocol to lessen the non-specific magnetic internalization during the magnet-assisted cell sorting (MACS) for HAMVECs purification.

Every single figure was finely chosen and well described. The supplementary information was well-detailed. For better clarity, I would suggest inserting schemes from the supplementary information in the main text (ex. SI4, SI5c, or SI6a).

Despite the high quality of the article, I would highlight one point: it is clear that the particle size has an impact on the IMPs internalization of adipose tissue-derived stromal/stem cells (ASCs). I am curious to know what is the optimal particle size for which the internalization becomes impossible. Also, bigger particles and a functionalization work could both lessen more the non-specific internalization. These points would bring a more precise insight into the technic you described since different size particles are already commercially available. I would suggest reading the following paper, it will probably give you more insights and ideas on that particular point. [Montel, L., Pinon, L., & Fattaccioli, J. (2019). A Multiparametric and High-Throughput Assay to Quantify the Influence of Target Size on Phagocytosis. *Biophysical Journal*, 1–12. <https://doi.org/10.1016/j.bpj.2019.06.021>].

Overall, the whole exposé was extremely clear and very convincing.

Reviewer #2 (Remarks to the Author):

This paper shows the application of a widely used technique called MACS (magnet assisted cell sorting) to isolate HAMVECs (human adipose tissue-derived microvascular ECs) from human fat. However, in that process, there is some contamination of adipose derived stromal cells (ASCs) which the authors propose that it could be mitigated by reducing the non-specific internalization of the immunomagnetic particles by the ASCs. Overall, the authors have done a number of interesting experiments, however, a few experiments maybe required to strengthen the claim.

Major comments

1) The claim (Page 3, last para) that the human fat samples require minimally invasive procedures to collect autologous endothelium for vascular tissue engineering applications is misleading because it is equivalent to a surgery and are usually collected as a byproduct of surgery. This process is no different from the commercially available HUVECs, which are isolated from discarded umbilical cord tissue.

In addition to this, the references to the statement "Moreover, the absence of an accessible alternative likely contributes to the sustained popularity of human umbilical vein ECs (HUVECs), the use of which continues to be reported in approximately 60% of publications amongst the biomaterials and tissue engineering communities despite being recognized as a clinically impractical source of endothelium" on page 4, first para identifies the potential reason for clinical impracticality of HUVECs as the heterogeneity of ECs which poses a concern for mimicking multiple regenerative processes in an organ/tissue-specific manner and limited life-span in culture. Neither of these problems are solved by isolation of HAMVECs with less than 100% purity, hence its significance as clinically relevant is not clear. It would have been clinically relevant if it was the isolation of stem cells or progenitors from the adipose tissue which can be trained to recapitulate organ-specific ECs.

2) On page 5, 2nd para ("The objective of this study was to develop an accessible, scalable, and

reliable method...tissue engineering") and again on page 14, last para. An experiment showing the scalability of the setup is required to claim this.

3) In the section ("Phenotypic assessment of HAMVECs implicated heterotypic cell-cell interactions in the modulation of their overgrowth by ASCs"), the morphological, molecular and functional assessments of other ECs with HAMVECs, it will also be required to confirm an important functional aspect of the enriched HAMVECs – vasculogenesis assay. It could be a simple in vitro or in vivo assay to confirm the functionality of HAMVECs by visualising vessel formation and that the ASC impurity doesn't interfere with HAMVEC's primary function. This is essential to show that the purity achieved is sufficient and no further enrichment techniques are required.

4) On page 8, first para at the end, there is a possibility that the isolated HAMVECs from human fat can also be endothelial progenitor cells as no specific elimination was done. An assay is recommended to show that the isolation of CD31+ HAMVECs are not adipose precursor cells. This could be done by incubating the enriched HAMVECs with adipogenic differentiation cocktail for several days. Similarly, the statement on page 18, first para, "...the acquisition of HAMVECs with the highest purity and least variability" would be addressed by this experiment.

5) On page 13, last para, last line, the authors mentioned that "...used to acquire HAMVECs in large quantities with high purities from five consecutive patients...". Whereas, on page 6, 2nd para, last line, the authors mention that only 3 of 20 patient samples were isolated with 98.0%-99.7% purity and remainder 17/20 with 85% purity using CD45-CD31+ MACS selection. And since, the authors mentioned using fig. 3j that purity of HAMVECs beyond 90% is essential to prevent the overgrowth of ASCs, which is achieved in the consecutive enrichment experiment, it is therefore, essential to show the data supporting the purities and quantities enriched for the remaining 17-5=12 samples which is not mentioned here, in order to strengthen the claim to be used as a reliable, less variable method.

Minor comments:

1) Page 19, in section Cell Isolation and Culture, number of patients not mentioned, should be 20.

Reviewer #3 (Remarks to the Author):

Antonyshyn et al developed a reliable method to improve tissue-derived microvascular endothelial cells (HAMVEC) acquisition from human fat by mitigating the non-specific uptake of immunomagnetic microparticles by the tissue-derived stromal/stem cells (ASC). First authors isolated HAMVEC using antiCD31 commercial immunomagnetic beads (IMP) selection and demonstrated that a purity of 90% is required to prevent ASC cells to overgrow HAMVEC in culture which was achieved only in a small fraction of attempt made (3/20). Interestingly, ASC fraction remained stable despite subsequent sequential enrichment for CD31 expression. They then investigated the phenotype of the isolated HAMVEC, comparing them to commercially available tissue-derived endothelial cells. The failure of sequential rounds of MACS enrichment to eliminate ASC was explained by the ability of ASC cells to bind and internalize IMP. A phenomenon that was independent on the IMP specificity but dependant on their size and the time of exposure. They then design cleavable cIMP beads in which antibody are attached to supermagnetic microparticles through a DNA linker that is cleaved by DNase treatment and show that HAMVEC purity is improved by the use of cIMP because of their larger size and their exclusion from culture after enrichment.

It is an important subject area, and builds on work from this group and others to find a reliable and autologous source of EC for vascular engineering. The experimental approach is valid, and the data is of high quality. Statistics tests are appropriate. Supplementary figures 4,5,6 and 8 are particularly helpful to understand the methodology. The abstract and summary are appropriate and clear.

There are several weaknesses that should be considered in the improvement of this manuscript.

1. Authors never show the number of HAMVEC isolated using their procedure. Since the HAMVEC as do other endothelial cells cannot be expanded indefinitely and their procedure will be in fine used to produce vascular tissue, It would be crucial to mention how many cells are obtained through their procedure and how many passages are required to obtain a reliable cell number for subsequent

analysis.

2. Line 102, Figure 1, authors show that HAMVEC comprised less than 1% of the total stromal vascular fraction while single-cell analysis (ref 33) suggested that endothelial cells represents 8% of total cells in adipose tissue. This suggests that a lot of endothelial cells are lost during the enzymatic digestion. Authors should clarify this discrepancy.

3. Related to comments 2, authors mention that the endothelial cells plasticity of the ASC was evident with LC-MS/MS detecting their expression of CD31, Ve-cadherin and vWF (Line 210). This doesn't necessary means that ASC can upregulate endothelial cells markers as stated by authors but could also reflect that a certain fraction of endothelial cells are not captured by their positive selection. This could be explained by two reasons : - the ratio of IMP/cells number is suboptimal (to many IMP or not enough) or more likely that the enzymatic digestion cleaves surface antigen from HAMVEC which are therefore not selected via their antiCD31-IMPs. This should be tested or at least discussed.

4. Figure 4 : all experiments are done in ASC culture, it will be crucial to demonstrate the presence and quantify the number of IMP in contaminating ASC cells from HAMVEC purified culture. This is crucial to evaluate in which degree the non-specific uptake of IMP is responsible for the lack of purity of HAMVEC selection.

5. Related to comment 4, DNase treatment is used in cell-sorting procedure to detach cells doublets and could have improved the HAMVEC purity simply by detaching ASC cells sticking to HAMVEC during the enrichment procedure. The effect of the DNase treatment should be tested on its own. Similarly, it is unclear what is the contribution of the size of the cIMP versus their ability to be excluded from culture in the greater purity obtained after 4 days of cells culture. This should be tested or stated clearly in the manuscript.

6. The mechanism by which ASC uptake IMP is not described, it will be interesting to test the kinetic of this phenomenon compares to the specific binding of antiCD31 IMP to HAMVEC during the 20min enrichment procedure. Can this non-specific uptake be mitigated by optimising the IMP/Cell number ratio or by favouring antigen-antibody interaction over cell/beads surface interaction?

7. Finally, I found Figure 6 particularly hard to understand and figures legends and results related to this figure should be rephrased to be as comprehensible as the rest of the paper. The use of DNase is not mentioned in the methods and makes it hard to understand when and how it was used.

Minor comments :

Figure 1f : Percent of cells should be indicated on this figure as in figure 1 a. b and c.

Line 71-74 introduction : Ref 13-14 should be acknowledged when methods using immunoselection for the enrichment of the endothelium are cited.

Reviewer #1 (Remarks to the Author):

Jeremy A. Antonyshyn and co-workers realized a very clear and helpful work for the biomedical and tissues engineering community. They worked on the isolation and enrichment of human adipose tissue-derived microvascular endothelial cells (HAMVECs), rare endothelium cells (almost 1% of the total stromal vascular fraction) for vascular tissue engineering purposes. The challenge addressed here is depicting a finely controlled protocol to lessen the non-specific magnetic internalization during the magnet-assisted cell sorting (MACS) for HAMVECs purification.

Reviewer's Comments:

1. Every single figure was finely chosen and well described. The supplementary information was well-detailed. For better clarity, I would suggest inserting schemes from the supplementary information in the main text (ex. SI4, SI5c, or SI6a).

Authors' Response: The experiment depicted in **Figure 9**, which demonstrated that mitigating the non-specific uptake of immunomagnetic microparticles (IMPs) by adipose tissue-derived stromal/stem cells (ASCs) facilitates the enrichment of human adipose tissue-derived microvascular endothelial cells (HAMVECs), contributed to some confusion during the peer review of this manuscript. Accordingly, a schematic has been included here, which depicts the magnet-assisted cell sorting (MACS) procedure used to acquire HAMVECs from enzymatically digested fat, as well as the different combinations of IMPs tested for the isolation and enrichment of HAMVECs using the *in vitro* model of their contaminated primary cultures (**Fig. 9a**).

2. Despite the high quality of the article, I would highlight one point: it is clear that the particle size has an impact on the IMPs internalization of adipose tissue-derived stromal/stem cells (ASCs). I am curious to know what is the optimal particle size for which the internalization becomes impossible. Also, bigger particles and a functionalization work could both lessen more the non-specific internalization. These points would bring a more precise insight into the technic you described since different size particles are already commercially available. I would suggest reading the following paper, it will probably give you more insights and ideas on that particular point. [Montel, L., Pinon, L., & Fattaccioli, J. (2019). A Multiparametric and High-Throughput Assay to Quantify the Influence of Target Size on Phagocytosis. *Biophysical Journal*, 1–12. <https://doi.org/10.1016/j.bpj.2019.06.021>].

Authors' Response: We thank the reviewer for their direction to the article and encouragement to further explore the effect of immunomagnetic microparticle (IMP) size on their uptake by adipose tissue-derived stromal/stem cells (ASCs). The results of this investigation are now included in the manuscript (**Fig. 8; Supplementary Fig. 9**; text on page 14, paragraph 2). Five distinct anti-CD31 IMPs ranging in modal diameter from $0.9 \pm 0.3 \mu\text{m}$ to $8.7 \pm 1.5 \mu\text{m}$ were investigated (**Fig. 8a; Supplementary Table 2**). There was an inverse relationship between IMP size and their uptake by ASCs (**Fig. 8b**; $p < 0.0001$). Interestingly, IMP size also had a significant effect on the temporal dynamics underlying their uptake (**Fig. 8b**; $p = 0.0003$). While the uptake of anti-CD31 IMPs $\geq 4.4 \mu\text{m}$ in diameter was exacerbated

in culture, those that were $\leq 3.9 \mu\text{m}$ in diameter were readily taken up by ASCs after 20 min in suspension and their culture did not significantly increase their uptake (**Fig. 8b**). This may be attributed to a greater capacity for ASCs to bind and retain smaller IMPs at the plasma membrane prior to their internalization (**Supplementary Fig. 9e**). These findings suggest that IMPs $\leq 3.9 \mu\text{m}$ in diameter cannot be used to isolate human adipose tissue-derived microvascular endothelial cells (HAMVECs) due to the capacity for ASCs to readily bind them during the labeling portion of the magnet-assisted cell sorting (MACS) procedure (i.e. 20 min in suspension), and that the non-specific uptake of anti-CD31 IMPs $\geq 4.4 \mu\text{m}$ in diameter can be mitigated by either limiting their exposure to the cells to 20 min in suspension or by using superparamagnetic microparticles of a larger diameter.

Three distinct anti-CD31 IMPs were included in this experiment in addition to those originally investigated, i.e. the commercial anti-CD31 IMPs and the anti-CD31 cIMPs. The greater polydispersity of these new IMPs (**Fig. 8a; Supplementary Table 2**) prompted us to report the sizes of all IMPs in terms of their modal, rather than mean, diameters. These changes have been made throughout the text and figures of the manuscript. Furthermore, and despite being comprised of the same materials (i.e. iron oxide-loaded polystyrene spheres), the three new anti-CD31 IMPs were not as autofluorescent as the commercial anti-CD31 IMPs and the anti-CD31 cIMPs originally investigated. Accordingly, the IMP uptake and localization assay had to be modified for this experiment, and the methodologies are delineated in the *Methods* (**Supplementary Fig. 9**; text on page 34, paragraph 3).

3. Overall, the whole exposé was extremely clear and very convincing.

Authors' Response: Thank you.

Reviewer #2 (Remarks to the Author):

This paper shows the application of a widely used technique called MACS (magnet assisted cell sorting) to isolate HAMVECs (human adipose tissue-derived microvascular ECs) from human fat. However, in that process, there is some contamination of adipose derived stromal cells (ASCs) which the authors propose that it could be mitigated by reducing the non-specific internalization of the immunomagnetic particles by the ASCs. Overall, the authors have done a number of interesting experiments, however, a few experiments maybe required to strengthen the claim.

Reviewer's Comments:

Major comments

1. The claim (Page 3, last para) that the human fat samples require minimally invasive procedures to collect autologous endothelium for vascular tissue engineering applications is misleading because it is equivalent to a surgery and are usually collected as a byproduct of surgery. This process is no different from the commercially available HUVECs, which are isolated from discarded umbilical cord tissue.

In addition to this, the references to the statement “Moreover, the absence of an accessible alternative likely contributes to the sustained popularity of human umbilical vein ECs (HUVECs), the use of which continues to be reported in approximately 60% of publications amongst the biomaterials and tissue engineering communities despite being recognized as a clinically impractical source of endothelium” on page 4, first para identifies the potential reason for clinical impracticality of HUVECs as the heterogeneity of ECs which poses a concern for mimicking multiple regenerative processes in an organ/tissue-specific manner and limited life-span in culture. Neither of these problems are solved by isolation of HAMVECs with less than 100% purity, hence its significance as clinically relevant is not clear. It would have been clinically relevant if it was the isolation of stem cells or progenitors from the adipose tissue which can be trained to recapitulate organ-specific ECs.

Authors' Response: The reviewer is correct in identifying that harvesting fat is a clinical procedure. However, fat can be harvested using a cannula coupled to a liposuction pump or even a syringe¹, meaning that human adipose tissue-derived microvascular endothelial cells (HAMVECs) can in fact be acquired using minimally invasive procedures. The text in the *Introduction* has been revised to better reflect this (text on page 4, paragraph 1, line 2) and is supported by a new reference that describes some of the procedures available to harvest fat¹.

It is important to understand that immunogenicity differentiates HAMVECs from human umbilical vein endothelial cells (HUVECs). HUVECs are allogeneic and, therefore, can be considered to be less clinically practical than HAMVECs, which can be harvested for autologous use using the aforementioned minimally invasive procedures¹. The importance of immunogenicity to the clinical translation of cellular therapies is recognized by leading clinical scientists². The text in the *Introduction* has been revised to better reflect this (text on page 4, paragraph 1, line 12) and is supported by a new reference that describes some of the

immunological considerations that challenge the clinical translation of allogeneic cell therapies². Accordingly, a readily accessible source of autologous endothelium may ultimately help facilitate the clinical translation of vascularized, tissue-engineered products.

The references to the statement, “Moreover, the absence of an accessible alternative likely contributes to the sustained popularity of human umbilical vein ECs (HUVECs), the use of which continues to be reported in approximately 60% of publications amongst the biomaterials and tissue engineering communities despite being recognized as a clinically impractical source of endothelium”,^{3,4} were intended to demonstrate the scope of use of HUVECs for vascular tissue engineering. The text in the *Introduction* has been revised to make this clear (text on page 4, paragraph 1, line 12).

Although efforts towards sourcing organ and tissue -specific endothelium for vascular tissue engineering is warranted, the field currently lacks a readily accessible source of autologous endothelium of any specification. Our investigation was meant to contribute knowledge towards addressing this gap. Notably, and as described in the *Discussion*, HAMVECs isolated on the basis of a conventional endothelial CD45⁻CD31⁺ immunophenotype are likely comprised of a heterogenous mixture of endothelial subpopulations, such as the arterial, venous, and lymphatic specifications. This hypothesis is supported by single cell RNA sequencing studies of fat⁵, and suggests that HAMVECs may be further enriched for desirable phenotypic specifications for tailored vascular tissue engineering applications – an important step towards recapitulating endothelial heterogeneity and a subject of ongoing investigation by our lab.

With regard to the limited life-span of primary endothelial cells in culture – this challenge is mitigated by the abundant and uniquely dispensable nature of fat¹ as described in the *Introduction and Discussion*. That is, the need for the culture-mediated expansion of HAMVECs can be reduced by simply harvesting more fat, albeit within the expendable limit presented by the patient. Accordingly, HAMVECs are not only autologous, but are also readily accessible in large quantities. These features underpin the clinical appeal of fat for vascular tissue engineering.

References

1. Gir P, Brown SA, Oni G, Kashefi N, Mojallal A, Rohrich RJ. Fat grafting: evidence-based review on autologous fat harvesting, processing, reinjection, and storage. *Plast Reconstr Surg* 130:249-258,2012.
2. Petrus-Reurer S, Romano M, Howlett S, Jones JL, Lombardi G, Saeb-Parsy K. Immunological considerations and challenges for regenerative cellular therapies. *Commun Biol* 4:798,2021.
3. Wang K, Lin RZ, Melero-Martin JM. Bioengineering human vascular networks: trends and directions in endothelial and perivascular cell sources. *Cell Mol Life Sci* 76:421–439,2019.

4. Hauser S, Jung F, Pietzsch J. Human endothelial cell models in biomaterial research. *Trends Biotechnol* 35:265–277,2017.
 5. Vijay J, Gauthier MF, Biswell RL, Louiselle DA, Johnston JJ, Cheung WA, *et al.* Single-cell analysis of human adipose tissue identifies depot- and disease-specific cell types. *Nat Metab* 2:97–109,2020.
2. On page 5, 2nd para (“The objective of this study was to develop an accessible, scalable, and reliable method...tissue engineering”) and again on page 14, last para. An experiment showing the scalability of the setup is required to claim this.

Authors’ Response: We agree with the reviewer that data was not presented to support the scalability of this magnet-assisted cell sorting procedure. This comment was intended to suggest a potential future direction for this work, aimed at scaling-up the extraction of endothelial cells from human fat for clinical use. The text of the manuscript has been revised accordingly – i.e. claims of ‘scalability’ have been removed from the *Abstract*, *Introduction*, and *Conclusion*, and is only mentioned in the *Discussion* as a potential area for future research.

3. In the section (“Phenotypic assessment of HAMVECs implicated heterotypic cell-cell interactions in the modulation of their overgrowth by ASCs”), the morphological, molecular and functional assessments of other ECs with HAMVECs, it will also be required to confirm an important functional aspect of the enriched HAMVECs – vasculogenesis assay. It could be a simple in vitro or in vivo assay to confirm the functionality of HAMVECs by visualising vessel formation and that the ASC impurity doesn’t interfere with HAMVEC’s primary function. This is essential to show that the purity achieved is sufficient and no further enrichment techniques are required.

Authors’ Response: We thank the reviewer for this observation and have introduced additional studies to address their points. Specifically, the capacity of human adipose tissue-derived microvascular endothelial cells (HAMVECs) to form capillary-like tubules in basement membrane extract was compared to that of the representative endothelial cell (EC) controls. HAMVECs exhibited an angiogenic capacity comparable to that of the EC controls (**Fig. 2e**), further supporting their functional endothelial phenotype. This data has been added to the *Results* (text on page 8, paragraph 2, line 11) and the underlying assay, adapted from Arnaoutova *et al.*¹, is described in the *Methods* (text on page 27, paragraph 3).

An additional experiment has been performed in order to demonstrate that the purity of HAMVECs achieved in this investigation is sufficient and requires no further enrichment in order to prevent their overgrowth by adipose tissue-derived stromal/stem cells (ASCs). Specifically, the effect of seeding purity on the temporal composition of cultures was investigated (**Fig. 5c**). In the manuscript we advocate that the threshold of purity required to prevent stromal cell overgrowth is $\geq 98\%$, not 90%. Although similar magnitudes of population growth inhibition in both HAMVECs and ASCs were observed with a 90% seeding purity (**Fig. 5b**), the longer population doubling time of HAMVECs (**Fig. 5a**) suggests that an enrichment efficacy $> 90\%$ is required in order for the absolute population

growth rate of HAMVECs to exceed that of ASCs – i.e. the threshold of purity beyond which stromal cell overgrowth is precluded. In fact, cultures of HAMVECs established with 90% purity consistently exhibited significant overgrowth by ASCs within 7 days, and HAMVECs were virtually undetectable in these cultures after 28 days (**Fig. 5c**). In contrast, our isolates ($98.6 \pm 0.9\%$ CD45⁻CD31⁺; range: 98.0% – 99.7% CD45⁻CD31⁺) could be maintained in culture for 28 days without a significant decline in purity (**Fig. 5c**). These findings underscore the importance of seeding purity to the protracted stability of HAMVEC cultures, and suggest that no further enrichment is needed in cultures established with purities $\geq 98\%$. This data has been included in the *Results* (text on page 10, paragraph 2, line 11).

The panels comprising **Figures 4 & 5** were re-organized in order to enhance clarity and continuity. Specifically, **Figures 5a & 5b** were transferred from **Figure 4** and combined with **Figure 5c**, such that **Figure 4** depicts the data related to the proteomic assessment of HAMVECs that suggested that their proliferation is suppressed by ASCs, and **Figure 5** depicts the data which demonstrated that heterotypic cell-cell interactions modulate the overgrowth of HAMVECs by ASCs. The methodologies underlying **Figure 5c** are delineated in the *Methods* (text on page 32, paragraph 1).

References

1. Arnaoutova I, Kleinman HK. In vitro angiogenesis: endothelial cell tube formation on gelled basement membrane extract. *Nat Protoc* 5:628-635,2010.
4. On page 8, first para at the end, there is a possibility that the isolated HAMVECs from human fat can also be endothelial progenitor cells as no specific elimination was done. An assay is recommended to show that the isolation of CD31⁺ HAMVECs are not adipose precursor cells. This could be done by incubating the enriched HAMVECs with adipogenic differentiation cocktail for several days. Similarly, the statement on page 18, first para, "...the acquisition of HAMVECs with the highest purity and least variability" would be addressed by this experiment.

Authors' Response: We have obliged the recommendation of the reviewer. By way of introduction, human adipose tissue-derived microvascular endothelial cells (HAMVECs) were isolated from fat on the basis of a CD45⁻CD31⁺ immunophenotype. This cell-surface protein signature is characteristic of differentiated endothelium and not endothelial progenitor cells, which are characterized by a VEGFR2⁺CD45⁻CD31⁻ immunophenotype¹⁻³. The endothelial phenotype of the CD45⁻CD31⁺ HAMVECs was validated through comparisons with representative endothelial cell controls (**Fig. 2**). That the CD45⁻CD31⁺ immunophenotype used to isolate HAMVECs is characteristic of differentiated endothelium has now been emphasized in the *Results* (text on page 6, paragraph 1, line 1).

The commitment of HAMVECs to the endothelial lineage has been evaluated using the reviewer's recommended assay. Specifically, the adipogenic plasticity of HAMVECs was compared to that of the adipose tissue-derived stromal/stem cells (ASCs; **Fig. 3**). While the culture of ASCs in adipogenic media increased the size of their lipid droplets (**Fig. 3a**), it did not significantly increase their total abundance of lipids (**Fig. 3b**). This discrepancy may be

attributed to differences in their cellular densities, as the growth of ASCs was halted at confluence in adipogenic media but not in endothelial media (**Fig. 3a**). Nevertheless, ASCs exhibited a significantly greater accumulation of lipids than HAMVECs regardless of the media in which they were cultured (**Fig. 3b**). Furthermore, the culture of HAMVECs in adipogenic media induced their death rather than adipogenesis (**Fig. 3a**), supporting that HAMVECs are fully differentiated and committed to the endothelial lineage. This data has been included in the manuscript (text on page 8, paragraph 3) and the underlying methodologies, adapted from Kraus *et al.*,⁴ are delineated in the *Methods* (text on page 27, paragraph 4).

References

1. Nishikawa SI, Nishikawa S, Hirashima M, Matsuyoshi N, Kodama H. Progressive lineage analysis by cell sorting and culture identifies FLK1⁺VE-cadherin⁺ cells at a diverging point of endothelial and hemopoietic lineages. *Development* 125:1747-1757,1998.
2. Hirashima M, Kataoka H, Nishikawa S, Matsuyoshi N, Nishikawa SI. Maturation of embryonic stem cells into endothelial cells in an in vitro model of vasculogenesis. *Blood* 93:1253-1263,1999.
3. Yamashita J, Itoh H, Hirashima M, Ogawa M, Nishikawa S, Yurugi T, *et al.* Flk1-positive cells derived from embryonic stem cells serve as vascular progenitor cells. *Nature* 408:92-96,2000.
4. Kraus NA, Ehebauer F, Zapp B, Rudolphi B, Kraus BJ, Fraus D. Quantitative assessment of adipocyte differentiation in cell culture. *Adipocyte* 5:351–358,2016.
5. On page 13, last para, last line, the authors mentioned that "...used to acquire HAMVECs in large quantities with high purities from five consecutive patients...". Whereas, on page 6, 2nd para, last line, the authors mention that only 3 of 20 patient samples were isolated with 98.0%-99.7% purity and remainder 17/20 with 85% purity using CD45-CD31+ MACS selection. And since, the authors mentioned using fig. 3j that purity of HAMVECs beyond 90% is essential to prevent the overgrowth of ASCs, which is achieved in the consecutive enrichment experiment, it is therefore, essential to show the data supporting the purities and quantities enriched for the remaining 17-5=12 samples which is not mentioned here, in order to strengthen the claim to be used as a reliable, less variable method.

Authors' Response: A total of 25 patients were enrolled in this study. Initially twenty patients were enrolled in order to successfully isolate human adipose tissue-derived microvascular endothelial cells (HAMVECs) from three different donors, and then an additional five patients were enrolled in order to validate the new magnet-assisted cell sorting (MACS) procedure. The text in the *Results* has been revised to make this clear (text on page 6, paragraph 2, line 6; page 6, paragraph 3; and, page 16, paragraph 1, line 10).

In the initial cohort of 20 patients, primary cultures were primarily evaluated by transmission light microscopy in order to assess their extent of stromal cell overgrowth. The purity of the three primary cultures that were identified to be virtually free of contaminating stromal cells by microscopy were assessed by flow cytometry. Of the remaining 17 cultures, only some were assessed by flow cytometry so as to identify the immunophenotype of the contaminating stromal cells for further investigation – not to characterize their purity. The text in the *Results* has been revised to make this clear (text on page 6, paragraph 2, line 6; page 6, paragraph 3), as well as the text in the caption of **Figure 1**.

The experiment depicted in **Figure 9** demonstrated that mitigating the non-specific uptake of immunomagnetic microparticles (IMPs) by adipose tissue-derived stromal/stem cells (ASCs) facilitates the enrichment of HAMVECs. The greater efficacy of these MACS strategies relative to the use of the commercial anti-CD31 IMPs was clearly demonstrated by this experiment. Moreover, the efficacy of the best MACS strategy identified by this experiment was validated by using it to successfully isolate HAMVECs from the additional five, and consecutive, patients. Notably, it yielded purities comparable to that observed in the penultimate experiment ($98.9 \pm 0.7\%$ vs. $98.7 \pm 0.5\%$, respectively), supporting the validity of the *in vitro* model.

Minor comments:

6. Page 19, in section Cell Isolation and Culture, number of patients not mentioned, should be 20.

Authors' Response: A total of 25 patients were enrolled in this study. Initially twenty patients were enrolled in order to successfully isolate human adipose tissue-derived microvascular endothelial cells (HAMVECs) from three different donors, and then an additional five patients were enrolled in order to validate the new magnet-assisted cell sorting (MACS) procedure. The total number of patients enrolled in this study has been described in the *Methods* (text on page 22, paragraph 1, line 1).

Reviewer #3 (Remarks to the Author):

Antonyshyn et al developed a reliable method to improve tissue-derived microvascular endothelial cells (HAMVEC) acquisition from human fat by mitigating the non-specific uptake of immunomagnetic microparticles by the tissue-derived stromal/stem cells (ASC). First authors isolated HAMVEC using antiCD31 commercial immunomagnetic beads (IMP) selection and demonstrated that a purity of 90% is required to prevent ASC cells to overgrow HAMVEC in culture which was achieved only in a small fraction of attempt made (3/20). Interestingly, ASC fraction remained stable despite subsequent sequential enrichment for CD31 expression. They then investigated the phenotype of the isolated HAMVEC, comparing them to commercially available tissue-derived endothelial cells. The failure of sequential rounds of MACS enrichment to eliminate ASC was explained by the ability of ASC cells to bind and internalize IMP. A phenomenon that was independent on the IMP specificity but dependant on their size and the time of exposure. They then design cleavable cIMP beads in which antibody are attached to supermagnetic microparticles through a DNA linker that is cleaved by DNase treatment and show that HAMVEC purity is improved by the use of cIMP because of their larger size and their exclusion from culture after enrichment.

It is an important subject area, and builds on work from this group and others to find a reliable and autologous source of EC for vascular engineering. The experimental approach is valid, and the data is of high quality. Statistics tests are appropriate. Supplementary figures 4,5,6 and 8 are particularly helpful to understand the methodology. The abstract and summary are appropriate and clear.

There are several weaknesses that should be considered in the improvement of this manuscript.

Reviewer's Comments:

1. Authors never show the number of HAMVEC isolated using their procedure. Since the HAMVEC as do other endothelial cells cannot be expanded indefinitely and their procedure will be in fine used to produce vascular tissue, It would be crucial to mention how many cells are obtained through their procedure and how many passages are required to obtain a reliable cell number for subsequent analysis.

Authors' Response: The yield of stromal vascular cells from enzymatically digested human subcutaneous abdominal white adipose tissue was $6.6 \pm 4.7 \times 10^5$ cells per gram of tissue. The yield of human adipose tissue-derived microvascular endothelial cells (HAMVECs) was $4.6 \pm 3.0 \times 10^3$ cells per gram, and that of adipose tissue-derived stromal/stem cells (ASCs) was $3.8 \pm 2.7 \times 10^5$ cells per gram. This data has been included in the text of the *Results* (text on page 5, paragraph 3, line 2; page 6, paragraph 2, line 3; and, page 7, paragraph 2, line 2).

As described in the *Introduction* and *Discussion*, the need for the culture-mediated expansion of HAMVECs is mitigated by the abundant and uniquely dispensable nature of adipose tissue. The number of passages needed depends on the amount of fat harvested and the number of HAMVECs needed for downstream applications. The data provided on yield, coupled with

the population doubling times (**Fig. 5a**), will allow investigators to estimate the need for the culture-mediated expansion of HAMVECs for their intended use.

2. Line 102, Figure 1, authors show that HAMVEC comprised less than 1% of the total stromal vascular fraction while single-cell analysis (ref 33) suggested that endothelial cells represents 8% of total cells in adipose tissue. This suggests that a lot of endothelial cells are lost during the enzymatic digestion. Authors should clarify this discrepancy.

Authors' Response: The prevalence of human adipose tissue-derived microvascular endothelial cells (HAMVECs) observed in this investigation ($0.9 \pm 0.6\%$; **Fig. 1a**) is comparable to the generally accepted prevalence of microvascular endothelial cells in tissues (1-2%)¹. This is now highlighted in the text of the *Results* (text on page 6, paragraph 2, line 2). Discrepancies in the literature may be predominantly attributed to different criteria used to define an endothelial cell (e.g. CD31⁺ vs. CD45⁻CD31⁺), although differences in the underlying patient populations, donor sites, and methodologies likely contribute to this variability as well.

References

1. van Beijnum JR, Rousch M, Castermans K, van der Linden E, Griffioen AW. Isolation of endothelial cells from fresh tissues. *Nat Protoc* 3:1085-1091,2008.
3. Related to comments 2, authors mention that the endothelial cells plasticity of the ASC was evident with LC-MS/MS detecting their expression of CD31, Ve-cadherin and vWF (Line 210). This doesn't necessary means that ASC can upregulate endothelial cells markers as stated by authors but could also reflect that a certain fraction of endothelial cells are not captured by their positive selection. This could be explained by two reasons : - the ratio of IMP/cells number is suboptimal (to many IMP or not enough) or more likely that the enzymatic digestion cleaves surface antigen from HAMVEC which are therefore not selected via their antiCD31-IMPs. This should be tested or at least discussed.

Authors' Response: We thank the reviewer for these important thoughts. We have considered them carefully and note that, as described in the *Results*, the capacity for adipose tissue-derived stromal/stem cells (ASCs) to express characteristic endothelial markers and functions is well established¹. In fact, it has previously misled many to believe that they can readily differentiate ASCs into fully functional endothelial cells (ECs)¹. The efficacy of the magnet-assisted cell sorting procedure in depleting leukocytes and ECs in order to acquire CD45⁻CD31⁻ ASCs was validated by flow cytometry prior to their proteomic assessment (**Fig. 1**). Furthermore, the limited expression of CD31 in ASCs detected by liquid chromatography tandem mass spectrometry was further validated by flow cytometry (**Fig. 7e**; **Supplementary Fig. 8**). Specifically, the slight shift in the median fluorescence intensity of the entire population, in contrast to a distinct CD31⁺ subpopulation, supports the limited endothelial plasticity of ASCs rather than the presence of residual ECs (**Fig. 7e**; **Supplementary Fig. 8**). While the enzymatic cleavage of CD31 is a remote possibility, it is refuted in the current work by the demonstration of the reproducible and successful isolation

of CD45⁻CD31⁺ HAMVECs and the validation of its sustained cell-surface localization by flow cytometry.

References

1. Antonyshyn JA, McFadden MJ, Gramolini AO, Hofer SOP, Santerre JP. Limited endothelial plasticity of mesenchymal stem cells revealed by quantitative phenotypic comparisons to representative endothelial cell controls. *Stem Cells Transl Med* 8:35-45,2019.
4. Figure 4 : all experiments are done in ASC culture, it will be crucial to demonstrate the presence and quantify the number of IMP in contaminating ASC cells from HAMVEC purified culture. This is crucial to evaluate in which degree the non-specific uptake of IMP is responsible for the lack of purity of HAMVEC selection.

Authors' Response: In the revised manuscript, **Figure 6** (previously **Fig. 4**) depicts data related to the uptake of the commercial anti-CD31 immunomagnetic microparticles (IMPs) by the adipose tissue-derived stromal/stem cells (ASCs). The contribution of this phenomenon to the poor efficacy of sequential enrichments of contaminated cultures of HAMVECs was assessed in the experiment depicted in **Figure 9**, which supported a role for the non-specific uptake of anti-CD31 IMPs by ASCs in undermining the efficacy of subsequent enrichments for HAMVECs (**Fig. 9b & 9d**). These points have now all been emphasized in the revised manuscript to improve clarity.

5. Related to comment 4, DNase treatment is used in cell-sorting procedure to detach cells doublets and could have improved the HAMVEC purity simply by detaching ASC cells sticking to HAMVEC during the enrichment procedure. The effect of the DNase treatment should be tested on its own. Similarly, it is unclear what is the contribution of the size of the cIMP versus their ability to be excluded from culture in the greater purity obtained after 4 days of cells culture. This should be tested or stated clearly in the manuscript.

Authors' Response: Again, we thank the reviewer for these thoughtful comments. While the aggregation of human adipose tissue-derived microvascular endothelial cells (HAMVECs) with adipose tissue-derived stromal/stem cells (ASCs) is a plausible hypothesis underlying the contamination of primary cultures of HAMVECs, which we had also previously considered, the experiment depicted in **Figure 9** demonstrates that the non-specific uptake of anti-CD31 immunomagnetic microparticles (IMPs) by ASCs is the predominant factor undermining the efficacy of sequential enrichments of HAMVECs. The comparative efficacies of increasing the size of the IMPs vs. excluding them from cultures all together are depicted in **Figure 9d**. The efficacy of the latter was further validated by successfully isolating HAMVECs from five consecutive patients, supporting that it is the non-specific uptake of anti-CD31 IMPs by ASCs, rather than their aggregation with HAMVECs, that undermines their acquisition.

6. The mechanism by which ASC uptake IMP is not described, it will be interesting to test the kinetic of this phenomenon compares to the specific binding of antiCD31 IMP to HAMVEC during the 20min enrichment procedure. Can this non-specific uptake be mitigated by optimising the IMP/Cell number ratio or by favouring antigen-antibody interaction over cell/beads surface interaction?

Authors' Response: Titrating the number of anti-CD31 immunomagnetic microparticles (IMPs) to approximate the number of human adipose tissue-derived microvascular endothelial cells (HAMVECs) in suspension may be an alternative strategy to mitigate their non-specific uptake by adipose tissue-derived stromal/stem cells (ASCs) as the antigen-antibody interactions between HAMVECs and the anti-CD31 IMPs are likely favoured over the non-specific membrane-IMP interactions between ASCs and the anti-CD31 IMPs. However, inter-patient variability in the prevalence of HAMVECs may undermine the reproducibility of this approach. Hence, a more universally applicable solution would be preferred. The evidence presented throughout this manuscript demonstrates that increasing the size of the IMPs or limiting their exposure to the cells to 20 min in suspension is effective in mitigating their non-specific uptake by ASCs to enable the reliable, facile, and robust acquisition of HAMVECs – i.e. the ultimate goal of this investigation. Accordingly, we did not undertake these additional studies which would have the potential to increase complexities by requiring titrations for each patient.

7. Finally, I found Figure 6 particularly hard to understand and figures legends and results related to this figure should be rephrased to be as comprehensible as the rest of the paper. The use of DNase is not mentioned in the methods and makes it hard to understand when and how it was used.

Authors' Response: We acknowledge the challenging nature of this experiment and the presentation of its resulting data. The reviewer's comment has prompted its revision, which we hope clarifies the work. **Figure 9** (previously **Fig. 6**) demonstrates that mitigating the non-specific uptake of immunomagnetic microparticles (IMPs) by adipose tissue-derived stromal/stem cells (ASCs) facilitates the enrichment of human adipose tissue-derived microvascular endothelial cells (HAMVECs). The figure and related text in the *Results* (text on page 14, paragraph 3) have been substantially revised to improve clarity. Moreover, a schematic has been included that depicts the magnet-assisted cell sorting (MACS) procedure used to acquire HAMVECs from enzymatically digested fat, as well as the different combinations of IMPs tested for the isolation and enrichment of HAMVECs using the *in vitro* model of their contaminated primary cultures (**Fig. 9a**). The *Methods* have also been revised to elaborate on the use of DNase to enzymatically exclude the cleavable (c)IMPs from primary cultures (text on page 23, paragraph 2, line 6).

Minor comments:

8. Figure 1f : Percent of cells should be indicated on this figure as in figure 1 a. b and c.

Authors' Response: **Figure 1f** presents a pseudocolour plot depicting the composition of a primary culture of human adipose tissue-derived microvascular endothelial cells (HAMVECs) contaminated with spindle-shaped, fibroblast-like stromal cells. Flow cytometry was used to identify the immunophenotype of these contaminating cells rather than to assess the purity of the contaminated cultures, which has now been made clear in the *Results* (text on page 6, paragraph 2, line 6; and, page 6, paragraph 3) as well as the text in the caption of **Figure 1**. Accordingly, the central tendency and variability are not shown, but the proportion of cells exhibiting CD45⁻CD31⁺ and CD45⁻CD31⁻ immunophenotypes in this representative sample have been included as requested.

9. Line 71-74 introduction : Ref 13-14 should be acknowledged when methods using immunoselection for the enrichment of the endothelium are cited.

Authors' Response: The magnet-assisted cell sorting procedure used in this investigation was not derived, nor adapted, from these references, so they were not acknowledged in the *Methods*. The manufacturers of the immunomagnetic microparticles utilized in this investigation are provided in the *Methods*, and we encourage interested parties to look-up the related product specifications and standard operating procedures.

Tables & Figures: New and Revised for the Manuscript and Supplementary Information

Figure 1. Primary cultures of human adipose tissue-derived microvascular endothelial cells (HAMVECs) are often overgrown by residual adipose tissue-derived stromal/stem cells (ASCs) from the magnet-assisted cell sorting (MACS) procedure. The stromal vascular fraction of enzymatically digested human subcutaneous abdominal white adipose tissue (**a**) was depleted of CD45⁺ leukocytes prior to positively selecting for CD31 expression to establish primary cultures of CD45⁻CD31⁺ HAMVECs (**b, c**). Their primary cultures were often overgrown by residual CD45⁻CD31⁻ ASCs from the MACS procedure despite sequential enrichments for CD31 expression (**f, g**), prompting the retention of ASCs for downstream studies (**d, e**). Shown are representative pseudocolour plots depicting the composition of the different populations of cells (**a, b, d, f**), as well as representative photomicrographs depicting their corresponding morphologies (**c, e, g**). Scale bars represent 200 μ m; and, values, mean \pm standard deviation. While the isolation of HAMVECs was initially attempted from twenty patients (N = 20), only three cultures were visibly free of stromal cell overgrowth (N = 3); in the other 17 patients, cultures were visibly overgrown by stromal cells within two weeks (N = 17). The composition of the different populations of cells was assessed in three patients in all but visibly contaminated primary cultures of HAMVECs (**a, b, d**; N = 3), in which flow cytometry was used to elucidate the identity of the contaminating stromal cells rather than to assess their purity (**f**; N = 1).

Figure 2. Human adipose tissue-derived microvascular endothelial cells (HAMVECs) exhibit morphological, molecular, and functional hallmarks of endothelium. HAMVECs were compared with endothelial cell (EC) controls representative of the predominant endothelial specializations, namely human umbilical vein ECs (HUVECs; macrovascular, venous endothelium), human coronary artery ECs (HCAECs; macrovascular, arterial endothelium), and human dermal microvascular ECs (HDMVECs; microvascular endothelium). **(a)** Cobblestone-like morphology of endothelium. Scale bars represent 200 μ m. **(b)** Abundance of transcripts encoding CD31, vascular endothelial (VE)-cadherin, and von Willebrand Factor (vWF). Glyceraldehyde-3-phosphate dehydrogenase (GAPDH) was used as a loading control. Dashed line depicts a mean difference of zero; and, dotted lines, the equivalence margin (δ) used for the two one-sided test for equivalence. Values represent the mean \pm 90% confidence interval; and, Cq, the quantification cycle. **(c)** Expression and localization of the corresponding endothelial proteins. Scale bars represent 25 μ m. **(d)** Internalization of acetylated low-density lipoprotein (AcLDL). Solid and dashed lines represent ECs cultured in the presence and absence of Alexa Fluor 488 - conjugated AcLDL, respectively. Values represent mean \pm standard deviation. **(e)** Capillary-like tubulogenesis by ECs. Scale bars represent 200 μ m. All experiments were performed in biological triplicate, using cells derived from three different donors (N = 3).

Figure 3. Adipogenic plasticity evident in adipose tissue-derived stromal/stem cells (ASCs), not human adipose tissue-derived microvascular endothelial cells (HAMVECs). HAMVECs and ASCs were cultured in adipogenic medium for 10 days before their accumulation of lipids was assessed by Oil Red O with hematoxylin counterstaining. Endothelial medium was used as control. Shown are representative photomicrographs depicting the accumulation of lipids by HAMVECs and ASCs (a), as well as a bar graph delineating its quantification (b). Hematoxylin stains nuclei blue/purple, and Oil Red O stains lipids red/orange. Scale bars represent 100 μ m. Values represent mean \pm standard deviation; and, *, $p < 0.05$. Experiments were performed in biological triplicate, using cells derived from three different donors (N = 3).

Figure 4. Proteomic assessment of human adipose tissue-derived microvascular endothelial cells (HAMVECs) suggests that their proliferation is suppressed by adipose tissue-derived stromal/stem cells (ASCs). **(a)** Workflow depicting the proteomic comparison of HAMVECs with endothelial cell (EC) controls representative of the predominant endothelial specializations, namely human umbilical vein ECs (HUVECs; macrovascular, venous endothelium), human coronary artery ECs (HCAECs; macrovascular, arterial endothelium), and human dermal microvascular ECs (HDMVECs; microvascular endothelium). **(b)** Unsupervised hierarchical clustering of the proteomes of the ECs derived from four different vascular beds. **(c)** Distribution of detected proteins amongst the different ECs. **(d)** Hierarchical clustering of the proteins detected in different abundances ($p < 0.05$) between EC types. **(e)** Biological pathways differentiating HAMVECs from all other types of ECs. Black bars highlight those related to proliferation; and, white bars, metabolism. **(f)** Population doubling times of the different ECs, determined from their exponential growth phase observed over 7 days of culture. Values represent mean \pm standard deviation; and, *, $p < 0.05$ compared with HAMVECs. **(g)** Purities of the different EC cultures. Values represent mean \pm standard deviation; and, *, $p < 0.05$ compared with HAMVECs. **(h)** Representative zones of inhibition surrounding residual ASCs in primary cultures of HAMVECs that were maintained at confluence for over three weeks. Scale bar represents 500 μm . All experiments were performed in biological triplicate, using cells derived from three different donors ($N = 3$).

Figure 5. Heterotypic cell-cell interactions modulate the overgrowth of human adipose tissue-derived microvascular endothelial cells (HAMVECs) by adipose tissue-derived stromal/stem cells (ASCs). **(a)** Population doubling times of HAMVECs and ASCs, determined from their exponential growth phase observed over 7 days of culture. * $p < 0.05$ compared with HAMVECs. **(b)** Effect of seeding purity on the population growth rates of HAMVECs and ASCs, observed over 4 days of culture. * $p < 0.05$ compared with HAMVECs; # $p < 0.05$ compared with 100%. **(c)** Effect of seeding purity on the composition of cultures maintained at confluence for up to 28 days. * $p < 0.05$ compared with 100% seeding purity at respective time-point; # $p < 0.05$ compared with purity at day 1. Values represent mean \pm standard deviation. All experiments were performed in biological triplicate, using cells derived from three different donors (N = 3).

Figure 8. Size affects the extent and temporal dynamics underlying the uptake of immunomagnetic microparticles (IMPs) by adipose tissue-derived stromal/stem cells (ASCs). **(a)** Size distributions of five distinct anti-CD31 IMPs. * $p < 0.05$ compared with all other IMPs. **(b)** Total uptake (i.e. membrane-bound and/or internalized) of the different anti-CD31 IMPs by ASCs after 20 min in suspension (i.e. labeling conditions for magnet-assisted cell sorting) and 48 hr in culture. Values represent mean \pm standard deviation; * $p < 0.05$; and, n.s., not significant. This experiment was performed in biological triplicate, using cells derived from three different donors ($N = 3$).

Figure 9. Mitigating the non-specific uptake of immunomagnetic microparticles (IMPs) by adipose tissue-derived stromal/stem cells (ASCs) facilitates the enrichment of human adipose tissue-derived microvascular endothelial cells (HAMVECs). **(a)** Schematic depicting the magnet-assisted cell sorting (MACS) procedure used to acquire HAMVECs from enzymatically digested fat, as well as the different combinations of IMPs tested for the isolation and enrichment of HAMVECs using an in vitro model of their contaminated primary cultures. Specifically, HAMVECs and ASCs were seeded in a 9:1 proportion with or without IMPs to recapitulate their potential exclusion from primary cultures, made possible by enzymatically cleaving the deoxyribonucleic acid linkers coupling the antibodies to the superparamagnetic microparticles (i.e. cleavable (c)IMPs). Their enrichment was performed after 4 days, and their purities (% CD31⁺) were evaluated after a total of 7 days. **(b)** Effects of alternative target antigens (CD31 vs. CD93), sizes (4.4 μm IMPs vs. 4.8 μm cIMPs), and exposures (introduced vs. excluded from cultures) of IMPs on the enrichment efficacy of the MACS procedure. Values represent mean. **(c)** Effect of the target antigen on the enrichment efficacy of the MACS procedure. ‘None’ depicts the distribution of purities of co-cultures that were not subjected to MACS after four days; ‘CD31’, those that were enriched using anti-CD31 IMPs or anti-CD31 cIMPs; and, ‘CD93’, those that were enriched using anti-CD93 cIMPs. **(d)** Effect of introducing IMPs into primary cultures on the efficacy of their sequential enrichment. ‘None’ depicts the distribution of purities of enriched co-cultures that were free of IMPs and cIMPs at the time of MACS; ‘4.4 μm IMPs’, those that were laden with the smaller anti-CD31 IMPs; and, ‘4.8 μm cIMPs’, those that were laden with the larger anti-CD31 cIMPs or anti-CD93 cIMPs. **p* < 0.05; n.s., not statistically significant. Dashed lines in the violin plots represent median; dotted lines, quartiles; and, horizontal solid lines, range. The experiment was performed in biological triplicate, using cells derived from three different donors (N = 3).

Supplementary Table 2. Size distributions of the immunomagnetic microparticles (IMPs). IMPs are labeled on the basis of their target antigen and modal diameter. Their size distributions were evaluated using a Coulter counter, collecting data for a modal count of 5,000. cIMP represents cleavable IMPs; Std Dev, standard deviation; and, PDI, polydispersity index.

IMP	N	Mean (μm)	Median (μm)	Mode (μm)	Std Dev (μm)	PDI
0.9 μm Anti-CD31 IMPs	249,500	1.26	1.22	0.93	0.34	0.071
3.9 μm Anti-CD31 IMPs	217,400	5.07	4.89	3.91	1.24	0.060
4.4 μm Anti-CD31 IMPs	59,312	4.59	4.52	4.44	0.61	0.018
4.8 μm Anti-CD31 cIMPs	62,669	4.95	4.89	4.81	0.56	0.013
4.8 μm Anti-CD93 cIMPs	60,948	4.96	4.89	4.84	0.54	0.012
8.7 μm Anti-CD31 IMPs	60,592	8.84	8.71	8.65	1.50	0.029

Supplementary Figure 9. Size affects the extent and temporal dynamics underlying the uptake of immunomagnetic microparticles (IMPs) by adipose tissue-derived stromal/stem cells (ASCs). (a) Size distributions of five distinct anti-CD31 IMPs. * $p < 0.05$ compared with all other IMPs. (b) Schematic depicting the IMP uptake assay. ASCs were exposed to the different anti-CD31 IMPs for 20 min in suspension or different durations in culture before using magnet-assisted cell sorting (MACS) to separate IMP⁺ ASCs from IMP⁻ ASCs. IMP⁺ ASCs were stained with a

membrane dye before being recombined with the IMP⁻ ASCs. The mixture was then stained with a membrane-impermeable secondary antibody directed against the binding moiety of the IMPs to discriminate between their intracellular and extracellular localization. The mixture was assessed by flow cytometry. (c) Gating strategy used to assess the prevalence of IMP uptake by ASCs and their localization by flow cytometry. FSC-A represents forward scatter area; SSC-A, side scatter area; FSC-H, forward scatter height; FMO, fluorescence minus one controls. (d) Total uptake (i.e. membrane-bound and/or internalized) of the different anti-CD31 IMPs by ASCs after 20 min in suspension (i.e. labeling conditions for MACS) and different durations in culture. Values represent mean \pm standard deviation; *p < 0.05 relative to 20 min in suspension; #p < 0.05 relative to all smaller IMPs at the same time-point; and, †p < 0.05 relative to IMPs with a modal diameter of 4.8 μ m and 8.7 μ m. (e) Prevalence of IMP localization after 20 min in suspension (i.e. labeling conditions for MACS) and different durations in culture. *p < 0.05 compared with respective localization in suspension; #p < 0.05 compared with membrane-bound localization at the same time-point. Experiments were performed in biological triplicate, using cells derived from three different donors (N = 3).

REVIEWERS' COMMENTS:

Reviewer #1 (Remarks to the Author):

The authors answered precisely the questions and correctly took into account the suggestions. Particularly, they achieved a relevant work concerning particle sizes and their effects on cells.

Reviewer #2 (Remarks to the Author):

I am happy with the revision and recommend publication.

Reviewer #3 (Remarks to the Author):

My queries have all been addressed. The replies to other points raised are also convincing. This is a well designed study and the paper appears ready for publication.

Reviewer #1 (Remarks to the Author):

The authors answered precisely the questions and correctly took into account the suggestions. Particularly, they achieved a relevant work concerning particle sizes and their effects on cells.

Authors' Response: Thank you.

Reviewer #2 (Remarks to the Author):

I am happy with the revision and recommend publication.

Authors' Response: Thank you.

Reviewer #3 (Remarks to the Author):

My queries have all been addressed. The replies to other points raised are also convincing. This is a well designed study and the paper appears ready for publication.

Authors' Response: Thank you.